# Tracking the contribution of inductive bias to individualised internal models

**Balázs Török**[1,2,3], **David G. Nagy**[1,4], **Mariann Kiss**[2,3], **Karolina Janacsek**[5,6‡], **Dezső Németh**[3,5,7‡], **Gergő Orbán**[1‡*]

**1** Department of Computational Sciences, Wigner Research Centre for Physics, Budapest, Hungary, **2** Department of Cognitive Science, Faculty of Natural Sciences, Budapest University of Technology and Economics, Műegyetem rkp. 3., H-1111 Budapest, Hungary, **3** Brain, Memory and Language Research Group, Institute of Cognitive Neuroscience and Psychology, Research Centre for Natural Sciences, Budapest, Hungary, **4** Institute of Physics, Eötvös Loránd University, Budapest, Hungary, **5** Institute of Psychology, ELTE Eötvös Loránd University, Budapest, Hungary, **6** Centre for Thinking and Learning, Institute for Lifecourse Development, School of Human Sciences, Faculty of Education, Health and Human Sciences, University of Greenwich, London, United Kingdom, **7** Lyon Neuroscience Research Center (CRNL), Université Claude Bernard Lyon 1, Lyon, France

‡ These authors share senior author on this work.
* orban.gergo@wigner.hu

**Data Availability Statement:** The code used for this paper is available at https://www.github.com/mzperix/asrt-beamsampling. The repository contains links to the experimental data as well as the data used to generate the figures.

## Abstract

Internal models capture the regularities of the environment and are central to understanding how humans adapt to environmental statistics. In general, the correct internal model is unknown to observers, instead they rely on an approximate model that is continually adapted throughout learning. However, experimenters assume an ideal observer model, which captures stimulus structure but ignores the diverging hypotheses that humans form during learning. We combine non-parametric Bayesian methods and probabilistic programming to infer rich and dynamic individualised internal models from response times. We demonstrate that the approach is capable of characterizing the discrepancy between the internal model maintained by individuals and the ideal observer model and to track the evolution of the contribution of the ideal observer model to the internal model throughout training. In particular, in an implicit visuomotor sequence learning task the identified discrepancy revealed an inductive bias that was consistent across individuals but varied in strength and persistence.

## Author summary

Instead of mapping stimuli directly to response, humans and other complex organisms are thought to maintain internal models of the environment. These internal models represent parts of the environment that are most relevant for deciding how to act in a given situation and therefore are key to explaining human behaviour. In behavioural experiments it is often assumed that the internal model in the subject's brain matches the true model that governs the experiment. However this assumption can be violated due to a variety of reasons, such as insufficient training. Furthermore, the deviation of the internal model from the true model is not uniform across individuals, and therefore it summarizes the

**Funding:** This research was supported by the National Brain Research Program (project 2017-1.2.1-NKP-2017-00002, D.N., G.O.); Hungarian Scientific Research Fund (NKFIH-OTKA K K125343, G.O.; NKFIH-OTKA K 128016, D.N., NKFIH-OTKA PD 124148, K.J.); Janos Bolyai Research Fellowship of the Hungarian Academy of Sciences (K.J.); IDEXLYON Fellowship of the University of Lyon as part of the Programme Investissements d'Avenir (ANR-16-IDEX-0005) (D.N). B.T. was supported by scholarship by Budapest University of Technology and Economics as well as by Mozaik Education Ltd. (Szeged, Hungary). The funders had no role in study design, data collection and analysis, decision to publish, or preparation of the manuscript.

**Competing interests:** The authors have declared that no competing interests exist.

subjective beliefs of humans. In this paper, we provide a method to reverse engineer the internal model for individual subjects by analysing trial by trial behavioural measurements such as reaction times. We then track and analyse these reverse engineered models over the course of the experiment to see how participants trade off between an early inductive bias towards Markovian dynamics and the model that reflects the evidence that humans accumulate during learning about the actual statistics of the stimuli.

## Introduction

Building internal models is key to acting efficiently in the environment [1–3]. Consider for example observing the surface of a swift river: understanding how ripples and intermittent smooth patches are shaped by underwater rocks and understanding the strength required for pulling the paddle to propel a raft in the desired direction helps to plan the route of our raft downstream. Internal models represent expectations of what is going to happen next, how objects and other people can be expected to behave (often termed intuitive physics and intuitive psychology), what is the state of unobserved parts of the environment and consequently what actions lead to desired outcomes.

An ideal observer maintains an internal model that perfectly reflects the properties of the environment and our observations. Assuming that humans maintain an ideal observer model has been instrumental to understand behavior in a wide array of situations [4–8]. However, limited experience with rafting and uncertainty about riverbed geometry introduces deviations between the ideal observer model and the internal model actually maintained by individuals. Indeed, deviations from the true model of the environment were key to accurately predict human judgements when they interacted with physical constructs [9]. Identifying potential deviations can be crucial since assuming an ideal observer model instead of the actual internal model can result in misinterpretation of the computations underlying human decisions [10]. Extensive experience with the environment contributes to closing the gap between the ideal observer model and the internal model but individual differences can persist due to variance in prior experience, learning strategies and a range of other factors [11–14]. Consequently, accurate prediction of behavior, especially in early stages of learning, is only possible if we can retrieve the actual subjective internal models.

Potential sources of the deviation between the ideal observer model and the maintained internal model has recently been the subject of intense research [15, 16]. Studies have demonstrated that learning novel and complex statistics can lead to systematic deviations from the ground truth model [17, 18]. Mismatch between the predictions of an ideal observer model and human behaviour has been shown to be a consequence of computations relying on an internal model that deviates from the ground truth rather than sub-optimal computations [10, 19, 20]. Insights on the reasons for such deviations come from theoretical considerations. In general, perfect knowledge of the ideal observer model can be challenged by the high task and stimulus statistics complexity or by the insufficiency of available information early during learning [21–24]. From a theoretical perspective, learning can be more efficient if observers not only rely on observations but recruit earlier knowledge as well. For instance, previous experience with sea kayaking can provide skills for dealing with surface features such as whirlpools or rapids, despite the fact that more regular and larger amplitude waves are characteristic of the sea. Relying on earlier knowledge can be phrased as an inductive bias since this might help the interpretation of the current stimulus but at the expense of potentially introducing

distortions [25]. In summary, characterising inductive biases is key to understanding how the actual internal model maintained by humans is related to the ideal observer model.

To identify internal models, a method is required that can perform efficient inference of a flexible class of possible internal models from behavior. Recent years have seen a number of studies where behavior was used to infer complex internal models [26–28]. These studies investigated internal models adapted to natural-like stimuli, in which case the ideal observer was not feasible to identify. We seek to investigate a scenario where the internal model is complex but the ideal observer model is well defined. Importantly, unlike [28], we aim to develop a tool that can efficiently infer subjective internal models, such that individual differences in learning curves can identify the evolution of the internal model as the participant learns about unfamiliar stimulus statistics. For this, we need a (i) highly expressive class of internal models and (ii) behavioural measurements that are highly informative about the internal model. We proceed by choosing an experimental paradigm that satisfies (ii). In an experiment where trials are governed by temporal dynamics and therefore individual trials are not independent, the sequence of behavioural measurements have information content that far exceeds that of an independent and identically distributed (i.i.d.) experimental setting. It has been extensively documented that participants do pick up temporal regularities in experiments with stochastic dynamics [29–31]. Furthermore, individuals show high variation in their initial assumptions [29, 32]. Relying on a paradigm which features inter-trial dependencies unknown to participants, we aim to reverse-engineer the newly formed dynamical internal models of individuals. In order to satisfy (i), we propose to use infinite Hidden Markov Models (iHMMs, [33], for a brief introduction please read S1 Appendix). To infer the structure and dynamics of the iHMM we adopt and extend the Cognitive Tomography (CT) framework [27]. The proposed Cognitive Tomography model combines iHMMs and the linear ascend to threshold with ergodic rate (LATER) model [34] to relate subjective probabilities of individuals to response time measurements on a trial-by-trial basis.

In this paper we set out to infer individualised dynamical internal models from response time data using the Cognitive Tomography principle. We use an implicit sequence learning paradigm in which the stimulus sequence is characterised by challenging statistics novel to participants.

We take a data-driven approach where the structure of the internal model is discovered through modelling the subtle statistical structure present in response times. We track the evolution of the internal model over multiple days and thus obtain individual learning curves that provide unique insight into the way internal models are acquired by learning. After introducing the CT framework, we validate that the model structure inferred by CT corresponds to the internal model of individuals by testing the generalization capability of the inferred model across tasks and stimulus statistics. After validating CT we use it to gain insights into learning by assessing how the inferred model relates to a stimulus statistics driven component, the ideal observer model. We track the contribution of the ideal observer model to the internal model by assessing the amount of variance in response time explained by the ideal observer model relative to the internal model inferred by CT. The residual variance in the CT predictions not explained by the ideal observer is identified with the inductive bias that humans use when learning the task. We attempt to break down the variances in response times into two independent components: the ideal observer model and an inductive bias. We show that the internal model inferred through CT can be reliably broken down into the contributions of the ideal observer model and a simple dynamical model, the so called Markov model. While the contribution of the Markov model varies across participants, it can consistently account for the dominant portion of the residual variance across all participants. Finally, by tracking the evolution of the contributions of the two models we show how the two models are traded off during

learning. While learning strategies and efficiency of learning varies considerably across individuals, a consistent trend can be identified over days, in which initial dominance of the Markov model is gradually taken over by the ideal observer, indicating that the Markov model is a general inductive bias for learning the temporal structure of the stimulus. Taken together, our findings demonstrate that complex internal models can be inferred from response time measurements. Furthermore, our results suggest a new perspective on how humans trade-off inductive biases and evidence over the course of learning and also provide new tools to measure such inductive biases.

## Results

In order to test how behavioural data from individuals can be used to infer a dynamical probabilistic variable internal model and assess the contribution of inductive biases to the internal model, we used an experimental paradigm that could fulfil a number of key desiderata. First, the paradigm relies on across-trial dependencies; second, as in everyday tasks, the state of the environment cannot be unambiguously determined from the observation of momentary stimuli; third, the structure of the task is new to participants; fourth, the complexity of the task is relatively high, i.e. an a priori unknown number of latent states determine the observations; fifth, behavioural measurements during task execution are continuous, which ensures that rich inferences can be made. In the alternating serial response time task (ASRT, [35]) a stimulus can appear at four locations of a computer screen and the sequence of locations (untold to participants) follows a pre-specified structure (Fig 1A). In odd trials, the stimulus follows a 4-element sequence, while in even trials the stimulus appears at random at any of the positions with equal probability independently of all other trials (Fig 1, Methods). Such stimuli precluded unambiguously determining the state of the task solely based on a single trial's observation. There are an additional 5 random trials at the beginning of each block. Participants are tasked to give fast and accurate manual responses through key presses corresponding to the locations of the stimuli. We collected response time measurements for sequences of stimuli organized into blocks of 85 trials. A session consisted of 25 blocks and the performance was tracked for 8 days with one session on each day, during which the same stimulus statistics governed the stimuli, followed by two additional sessions on later days where the statistics of stimuli was altered (sessions were weekly spaced when possible, on occasions 2–3 day shifts were in place due to participant availability, S1 Fig).

We used the response times of individuals to infer a dynamical probabilistic latent variable model underlying their behaviour (Fig 1, Methods). We invoked the concept of CT to infer the internal model from a limited amount of data. CT requires the formulation of the generative model of the data, i.e. the process that produces behavioral data from observations. CT distinguishes two components of the model (Fig 1B, *blue boxes*): the internal model, which summarizes an individual's knowledge about the stimulus statistics and the behavioral model, which describes how behavioral responses are related to the internal model during the task that is being performed. Inference of the internal model requires inference how latent states evolve and how these determine the stimuli. By knowing the dynamics of latent states we can make predictions for the upcoming stimuli by establishing the subjective probability of possible subsequent elements of the stimuli. The behavioral model establishes how subjective probabilities of the internal model are related to behavioral outcome, which is the response time in our case. We used the LATER model to predict response times from subjective probabilities [34, 36]. The experimenter uses the observed data (Fig 1B, *grey boxes*), the stimulus sequence and response times, for the inference. The resulting CT model (S2 Fig) is implemented as a probabilistic program with components implemented in Stan [37].

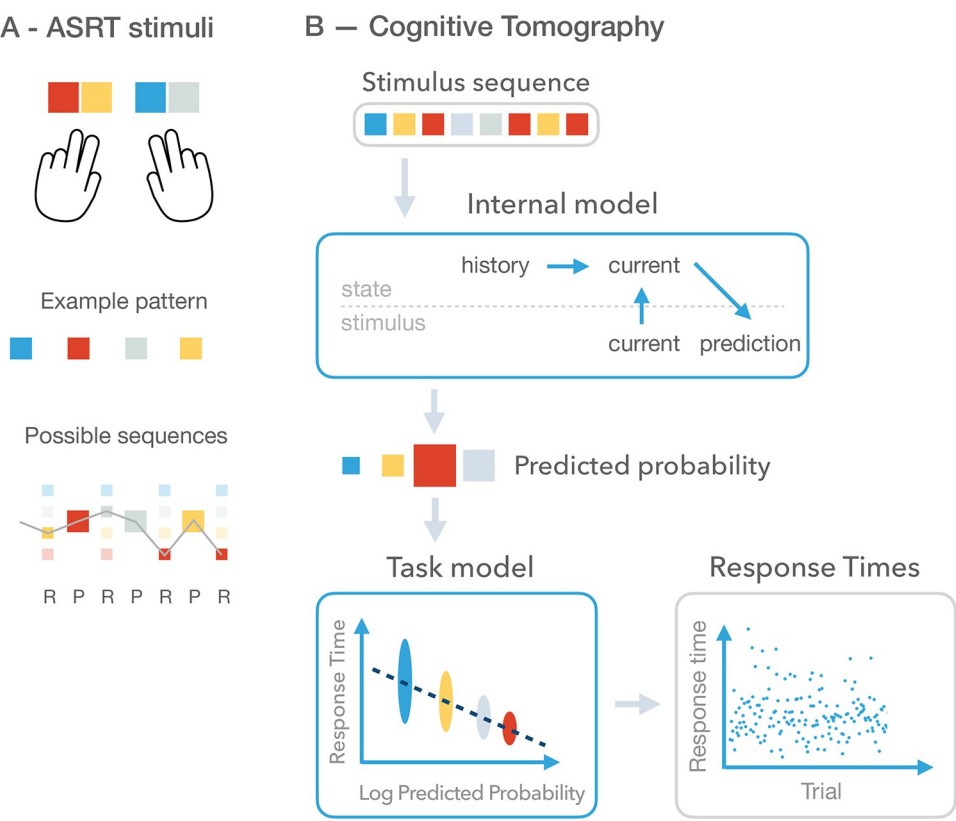

**Fig 1. Experimental paradigm and Cognitive Tomography (CT). A** *Top*: Behavioural responses: participants are responding with key presses on a keyboard where stimulus identities (shown as different *coloured squares*) are associated with unique keys. *Middle*: An example deterministic pattern sequence, which recurrently occurs in the stimulus sequence of a particular participant. Different participants are presented with permutations of this four-element sequence. *Bottom*: In the actual stimulus sequence presented to participants, the deterministic pattern sequence is interleaved with random items (*small squares*. Random items can be any of the four stimuli and can occur with equal probability (*size* of the square is proportional to the probability of a stimulus). *Grey line* indicates one particular realization of the stochastic sequence. **B** The probabilistic generative model underlying Cognitive tomography. The generative model describes the process how a stimulus sequence (*top grey box*) results in a behavioural response. A participant is assumed to use the internal model *top blue box* to make a prediction for the upcoming stimulus. The internal model assumes dynamics over the latent states. The current latent state is determined jointly by earlier states and the current observation. Based on the current latent state a prediction can be made on the probability of possible upcoming stimuli. The predicted probability (*size of squares* corresponds to the probability of prediction) is related to the behaviour through a behavioral model (*bottom blue box*). The behavioral model depends on the task being performed and therefore the type of response being predicted. Here, the logarithm of the predictive probability is mapped to a mean response time and actual response times are assumed to be noisy versions of this mean. Response times (*bottom grey box*) shown here are 400 trials from an example participant. Cognitive tomography uses the stimulus sequence and the sequence of behavioural responses (*grey boxes*) to infer the components of CT, the internal model and the behavioral model (*blue boxes*).

The iHMM model provides a flexible model class to infer latent variable models [33]. Similar to the classical Hidden Markov Model, learning entails the specification of transition probabilities between latent states along with the probability distributions of observations given a particular latent state (Fig 2A). Additional flexibility of iHMM is provided by not fixing the number of latent states but inferring this from data. This is implemented as a non-parametric Bayesian model (for a brief introduction into iHMM see S1 Appendix). In an iHMM, participants filter the information gained from the observations over time to estimate the possible latent state of the system (Fig 2Ba, *filled purple circles*). That is, they infer what history of events

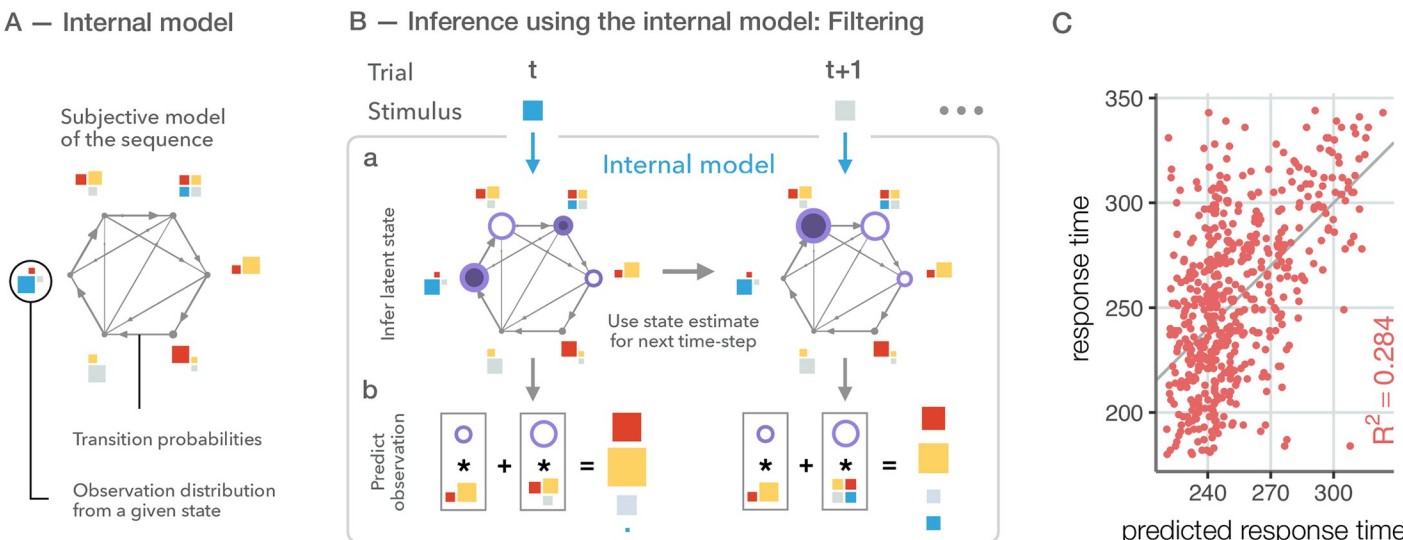

**Fig 2. Inference and predictions using the internal model. A** We formulate the internal model as an iHMM, where the number of latent states (*grey circles*), transitions between the states (*arrows*), and the distribution of possible stimuli for any given state (*coloured squares*) needs to be inferred by the experimenter. Width of arrows is proportional to transition probability and arrows are pruned if the transition probability is below a threshold; size of dots indicates the probability of self-transition. Size of stimuli is proportional to appearance probability in the given state. The result of inference is a distribution over possible model structures, the figure represents a single sample from such a distribution. **B** Evolving the internal model from trial $t$ to trial $t + 1$. At time $t$, participants use the internal model components to update their beliefs over the current state of the latent states (**Ba**, size of *dark purple discs* represent the posterior belief of the latent state based on the current observation, *blue square*). Then, participants play the model forward into the future (*open purple circles*). Finally, they generate predictions for the upcoming stimulus (**Bb**, *squares in grey boxes*) by summing over the possible future states (*open purple circles in grey boxes*). Participants use previous state beliefs and the new stimulus to update latent state beliefs. In this particular example, at trial $t + 1$ only one of the possible states can generate the observation, hence there is only one dark purple disk. Again, they play the dynamics forward and predict the next stimulus. **C** Predicted response times against actual response times are shown for individual trials for an example participant (*dots*). After training our inference algorithm on a training dataset of 10 blocks, we predict response times of another 10 blocks on the same day. Performance is measured as the trial-by-trial coefficient of determination between measured and predicted response times ($R^2$, *coloured label*).

could best explain the sequence of their stochastic observations. Then, they use their dynamical model to play the latent state forward (Fig 2Ba, *open purple circles*) and predict the next stimulus (Fig 2Bb).

During the eight days of exposure to the ASRT task participants undergo learning, which leads to a substantial reorganization of the internal model. Learning can be present on short (within day) or longer time scales. In our analysis we aimed at tracking the *across-day changes* in the internal model of individuals. The rationale behind this choice is twofold. First, while the non-parametric Bayesian approach is relatively data-thrifty, flexibility of the model comes at a price that it is still characterized by a larger number of parameters (a transition matrix with $N \cdot (N + 1)$ parameters and emission matrix with $4N$ parameters, where $N$ is the number of latent states). As a result, changes in the internal model cannot be reliably captured by a few button presses. In order to have a cross-validated measure of model performance we use non-overlapping data sets for learning the model and testing it. This also imposes a limit to how finely we can track changes in the internal model. Consequently, while theoretically there will be changes on a smaller time scale (especially on day one of the exposure), for practical reasons, to have a stable inference, we learn the model from the response times once in every session. Second, our analysis showed that there are substantial changes in the internal model even days after first exposure, which suggests slower learning processes, which can be reliably captured with across-day comparisons.

To test that the proposed inference algorithm is capable of the retrieval of the probabilistic model underlying response time sequences, we validated our inference algorithm on synthetic

data ([Methods], [S3 Fig]). We used three different model structures for validation, which were HMMs inferred from three different-length stimulus sequences (one sample from the iHMM inference in [[33]]). Similar to our human experiment data, we assessed CT by computing its predictive performance on synthetic response times. Further, since synthetic participants provide access to true subjective probabilities we also calculated performance on the ground truth subjective probabilities. We showed that the subjective probabilities can be accurately recovered from response times. As shown on [S3(D) Fig], standard deviations of participants' response times are within the range of successful model recovery.

To infer an internal model from response times, we inferred the internal model along with the parameters of the response time model on 10 blocks of trials measured at the second half of the session. Individual differences in internal models was captured by inferring internal models for every participant separately. We inferred the internal model from a single set of 10 blocks, once in a session. To check the validity of our response time model, we validated its basic assumptions. The response time model assumes that variance in response times comes from the joint effect of the variance in log predictive probabilities and an inverse Gaussian noise corrupting the subjective probabilities. If the fit of the internal and the response models are appropriate, the the residual variance, i.e. the variance not accounted for by the variance in the subjective probabilities predicted by the CT model, is expected to be inversely normally distributed. We checked this on the CT model on a subject by subject basis by contrasting the expected cumulative distribution of residuals with the measured cumulative distribution. This analysis demonstrated that residuals are close to a normal distribution ([S4 Fig]) with a single subject apparently having a bimodal residual distribution, potentially indicating additional structure in the internal model not captured by CT. Note, that throughout the analysis trials with fast response times are discarded (see [Methods] for details).

Response times could be predicted by CT efficiently even for individual trials as shown by the analysis of the response times from a single participant ($R^2(550) = 0.284$, $p < 0.001$, [Fig 2C]). The predicted distribution of response times closely matched that of the empirical distribution of response times (for an example, see [S5 Fig]). It is important to note that the predictive power was substantially increased by averaging over trials in the same positions of the sequence ([S6 Fig]). Despite the significant advantage of trial-averaged predictions, we believe that single trial predictions provide a more rigorous and important characterization of human behaviour therefore we evaluate model performances on an individual trial basis in the rest of the paper.

## Alternative models

Whether and how much the inferred internal model reflects the structure of the environment can be tested by contrasting the inferred CT model with the ideal observer model. Since we have full control over the generating process of the sequence of stimuli, the ideal observer model is identified with a generative model that has complete knowledge about the stimulus statistics and the only form of uncertainty afflicting inference stems from the ambiguity in the interpretation of observations rather than uncertainty in model structure or parameters. Assessment of the deviation of the CT and the ideal observer models can reveal the richness of the strategies pursued by humans when exposed to unfamiliar artificial stimulus statistics. Fixed parameters of the ideal observer also ensured that the changing task performance of humans could be directly compared across the course of learning to the same baseline. Importantly, 'learning' by participants during extended exposure throughout the experiment does not necessarily mean that their internal model gets gradually closer to the ideal observer model since even when more evidence is provided towards the true underlying model one can

commit more and more to a superstitious model. As a consequence, deviation can temporarily accumulate before converging towards the true model, resulting in nonlinear learning trajectories. The ideal observer model is the one that perfectly corresponds to the task structure. Importantly, this ideal observer model is part of the set of models that iHMM can learn. This internal model corresponds to a graphical model in which eight states are present representing the four alternating pattern and random states, such that pattern states are characterized by a single possible stimulus and random states are characterized by equal probability stimuli (Fig 3B). Thus, the ideal observer models bear strong similarities with the CT model but differ conceptually: the ideal observer model parameters are determined by stimulus statistics, while CT structure and parameters are determined by behavioral data.

A defining characteristic of the CT model was that it could model an arbitrarily rich temporal dependence between subsequent scenes by using latent states. A model that can only account for direct dependencies between consecutive scenes is the Markov model, which lacks the capability to represent latent states. This model learns the transition probabilities directly between observations, which is a simple but feasible model that can account for a wide range of everyday observations. Importantly, a Markov-like dynamics is a special case of the internals models that CT can represent.

Finally, the gold-standard for characterizing learning in an ASRT task is the so-called triplet model that tests the correlation of response times with the summary statistics of the stimulus sequence. We formulated this model as a trigram model. We include this model as well to

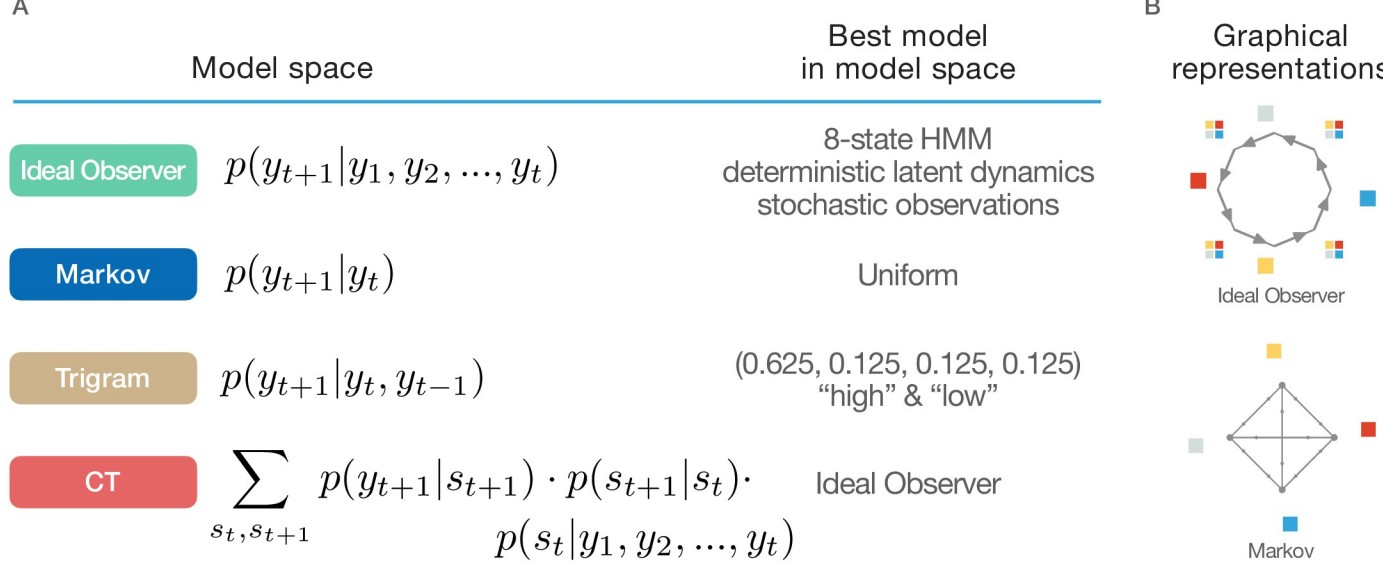

**Fig 3. Alternative models. A** Table of models and the maximum likelihood parameter sets for the stimuli in our experiment. The ideal observer model (the true generative model of the stimuli) can be formalized as an 8-state HMM with states Pattern1, Random1, Pattern2, Random2, Pattern3, Random3, Pattern4, Random4 where the pattern states produce the corresponding sequence element with probability 1 and all the random states produce any of the four observations with equal probability independently. The Markov model (where predictions are produced by conditioning only on the previous observation) fits the observations best when it predicts all observations with equal probability, since the marginal probabilities of any one stimulus is equal regardless what the previous observation was, because every other trial is random. The trigram model produces a "high triplet" prediction, where the next stimulus is the successor of the stimulus two trials ago in the pattern sequence (the current observation is either a random or a pattern element, each with 50% probability, with conditional probabilities of 100% or 25%, respectively). All alternatives have equal probability of 0.125. Note that the exact probabilities in this case are not relevant since the trials are categorized into two groups (high and low) and therefore the parameters of the response time model and these probabilities are underspecified. The CT model produces a prediction for the next stimulus via filtering. A latent state of the sequence is estimated from previous observations using a Hidden Markov Model. This flexible model space includes the ideal observer model as well as the Markov model as special cases. **B** Structure of the ideal observer model (*top panel*) and that of the Markov model (*bottom panel*). For the description of the graphical elements as Fig 2A.

compare its predictive performance with alternative models. Since the triplet model reflects the summary statistics of the stimulus this model bears resemblance to the ideal observer model albeit without assuming the ability to perform real inference in the model.

Direct comparison of alternative models is presented in Fig 3A (see also Methods).

## Comparison of ideal observer and CT performance to predict trial-by-trial behavior

We tracked the predictive performance of the ideal observer model through the eight days of exposure to a fixed stimulus statistics in the ASRT task. The ideal observer is determined by the stimulus structure therefore capturing across-individual differences is limited to different nuisance parameters, not characteristic of the internal model. Participant-averaged predictive performance of the ideal observer was not significantly above zero on the first day of training (one-sided $t(24) = -0.7692$, $p = 0.775$, $CI = [-0.0214, Inf]$, $d = 0.154$). Participant-averaged predictive performance was constantly increasing with the length of exposure, indicating that participants gradually acquire an internal model that accommodates the statistics of stimuli (Fig 4A).

Comparison of the internal model captured by CT to the ideal observer model reveals a consistent gap in predictive performance (Fig 4A). CT systematically outperformed the ideal observer on all eight days of exposure ($r(198) = 0.923$, $p < 0.001$) also demonstrating above chance predictive performance on the first day. The advantage of the CT model over the ideal observer was very consistent across participants as demonstrated by the participant-by-participant comparison of predictive performances on the eighth day of exposure (binomial test on MSE values 0.96, $n = 25$, $p < 0.001$, Fig 4B). In summary, while the ideal observer model demonstrates clear evidence that participants do gradually learn the stimulus statistics, CT reveals structure in responses that is not accounted for by the ideal observer model.

## Validation of the internal model

To verify that better predictive performance of the model identified by CT is not only a consequence of a more flexible model but is a signature of inferring a model that reflects better the

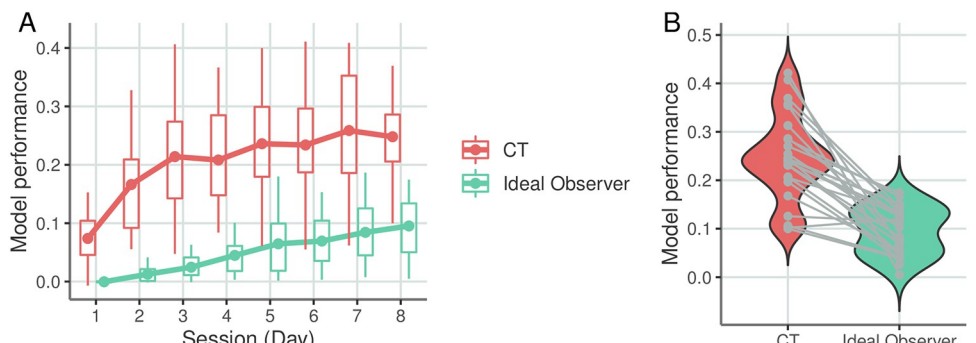

**Fig 4. Contrasting the ideal observer and CT performance in predicting trial-by-trial response times. A**, Performance of the two models in predicting response times on the eight days of exposure to the stimulus sequences governed by the same statistics. Performance is measured as the amount of variance in response times ($R^2$) explained by the particular model. Dots represent mean performance, boxes represent the 25 and 75 percentile of the performances across the population of 25 participants. **B**, Violin plot of the distribution of mode l performances across the participants on the eighth day of exposure. Grey dots indicate individual participants, lines connect model performances for the same participant. All data on the figure are cross-validated by fitting the model on a set of blocks late in the session and tested on non-overlapping earlier blocks.

properties of the internal model maintained by the participants, we perform two additional analyses. The core principle of Cognitive Tomography is to distinguish an internal model that captures a task-independent understanding of the statistical structure of the environment and a response model which describes task-specific behavioural responses. The validity of this principle can be assessed by manipulating the internal and behavioral models independently. First, we tested if the same internal model can be used to predict behavioral performance in a different task, that is, to predict behavioral measures different from those that the model was trained on. CT was trained on response times when participants pressed the correct key and here we replaced this task of predicting response times with the prediction of behavior in error trials, that is, in trials when the participant pressed the incorrect button. In particular, we aimed at predicting the trials in which a participant is likely to commit errors (because the subjective probability of the correct choice is relatively low) and also the erroneous response when an error occurs (the subjective probability of the choice relative to other potential choices). Second, we tested the usage of the internal model when stimulus statistics is manipulated. After completing eight days of training, participants were exposed to novel stimulus statistics and we tested if participants recruited the learned internal model only when the stimulus statistics matched the one the internal model had been learned on.

In error prediction we separated trials based on whether the participant pressed the key corresponding to the actual stimulus or any other keys. Note, that the internal models of CT were inferred only on correct trials using the response time model. We investigated two relevant hypotheses. First, a participant will more likely commit an error when their subjective probability of the stimulus is low. Second, when committing an error, their response will be biased towards their expectations. For reference, we contrasted the predictive performance of CT with the ideal observer model. We compared the rank of the subjective probability of the upcoming stimulus both for correct and incorrect trials (Fig 5A). CT ranked highest the upcoming stimulus in correct trials above chance (0.461, $n = 18473$, $p < 0.001$) and significantly below chance for incorrect trials (0.175, $n = 2777$, $p < 0.001$). Ideal observer model excelled at predicting the correct responses, as it ranked the correct responses high above chance (0.635, $n = 18473$, $p < 0.001$). However, it also assigned the highest probability to the upcoming stimulus in incorrect trials (0.315, $n = 2777$, $p < 0.001$). Ranking of incorrect responses was above chance for both models (Fig 5B).

We obtained a participant-by-participant assessment of the difference between model performances in predicting error trials by calculating ROC curves of the models based on the subjective probabilities assigned to upcoming stimuli (Fig 5C and S7 Fig). Area between two ROC curves characterizes the performance difference between models and CT is shown to consistently outperform the ideal observer model in distinguishing correct choices from incorrect choices (paired t-test on AUC values one-sided $t(24) = 6.185$, $p < 0.001$, $CI = [0.033, Inf]$, $d = 1.1$, Fig 5D). Thus, CT can perform across-task predictions and it substantially outperforms the ideal observer model as well.

We also tested the hypothesis whether participants use a single model to represent the sequence or they are capable of holding multiple models and recruiting them appropriately [38–40]. In particular, when we changed the underlying pattern sequence in the task, we expected participants to start learning a new model instead recalibrating the same model used in the first eight days. There are two major pieces of evidence at hand. Firstly, as expected, the internal model inferred on Day 8 does significantly worse in predicting behaviour when a new sequence is present on Day 9 (one-sided $t(24) = 4.958$, $p < 0.001$, $CI = [0.0746, Inf]$, $d = 1.06$). Similarly, the internal model inferred on Day 9 predicts human behaviour significantly worse on Day 8 (one-sided $t(24) = 4.9$, $p < 0.001$, $CI = [0.0616, Inf]$, $d = 0.963$). The real test to using two models is on Day 10, when the two underlying sequences are alternating every five blocks,

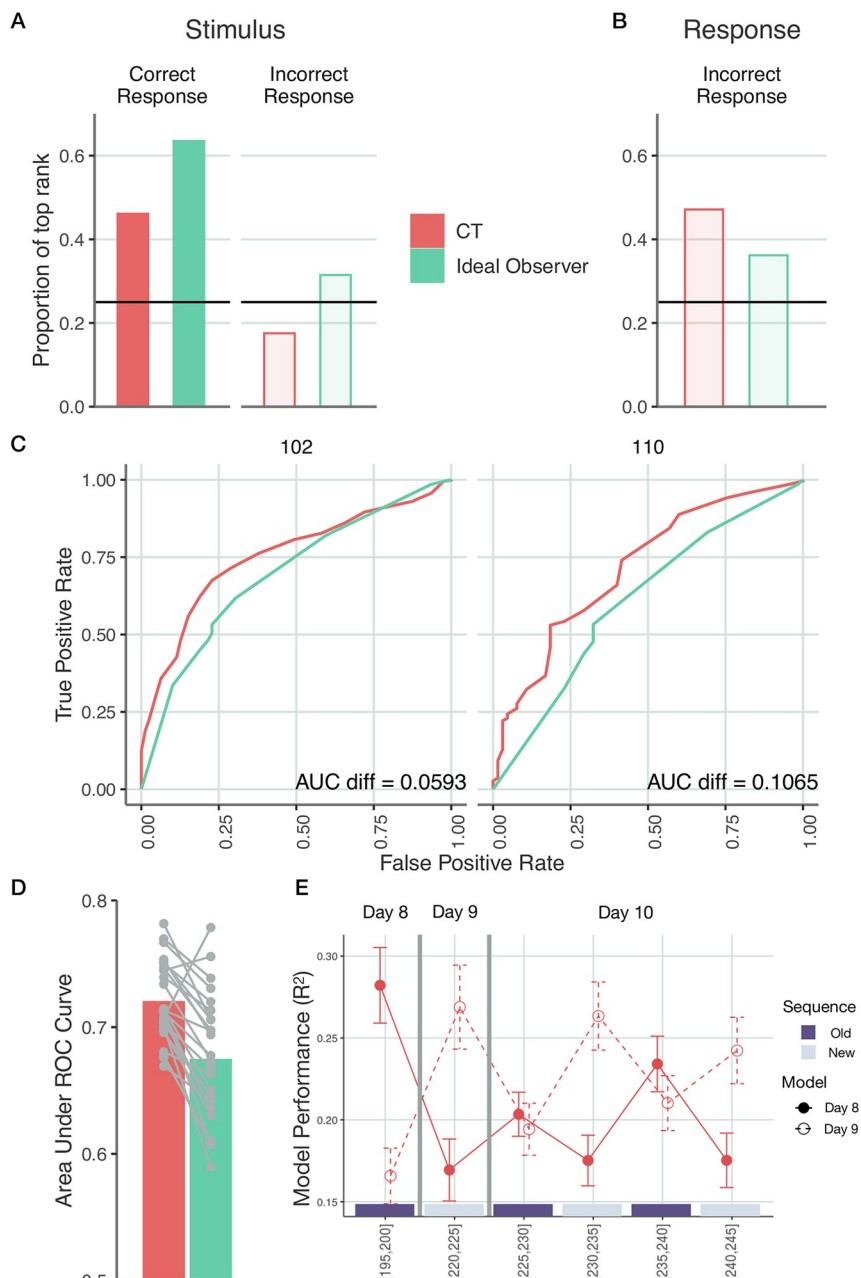

**Fig 5. Validation of the inferred internal model by selectively changing the task and the stimulus statistics. A-D** Choice predictions by CT (*red*) and the ideal observer model (*green*). Models are trained on response times for correct key presses on Day 8 and tested on both correct and error trials the same day. **A**, Proportion of trials where the model ranked the upcoming stimulus first. For correct trials both models have preference for the stimulus. For incorrect trials, the ideal observer model falsely predicts the stimulus in more than a quarter of the time. **B**, Proportion of trials where the model ranked the button pressed by the participant first. For incorrect responses, both models display a preference towards the actually pressed key over alternatives. **C**, ROC curves for two example participants based on the subjective probabilities of upcoming stimuli (held-out dataset). Area under the ROC curve characterizes the performance of a particular model in predicting error trials. **D**, Area under ROC curve. Grey dots show individuals, bars show means. **E**, Investigating new internal models that emerge when new stimulus sequences are presented. Participant-averaged performance of predicting response times on Day 8–10 using CT-inferred models that were trained on Day 8 (*filled red symbols*) and Day 9 (*open red symbols*) on stimulus sequences governed by Day 8 or Day 9 statistics. On Day 9 a new stimulus sequence was introduced, therefore across-day prediction of response times corresponded to across sequence predictions. Training of the models was performed on 10 blocks of trials starting from the 11th block and prediction was performed on the last five blocks of trials (the index of the blocks used in

testing is indicated in brackets). On Day 10, stimulus sequence was switched in 5-block segments between sequences used during Day 8 and Day 9 (*purple and grey bars* indicate the identity of stimulus sequence with colours matching the bars used in Day 8 and Day 9. Error bars show 2 s.e.m. over participants. Stars denote $p < 0.05$ difference.

starting with Day 8 sequence in blocks 1–5. Specificity of response time statistics to the stimulus statistics is tested by predicting Day 10 performance using Day 8 and Day 9 models. The Day 8 model more successfully predicts response times in blocks relying on Day 8 statistics than on blocks with Day 9 statistics (one-sided $t(24) = 3.734$, $p < 0.001$, $CI = [0.0236, Inf]$, $d = 0.594$) and the opposite is true for the Day 9 model (one-sided $t(24) = 3.528$, $p < 0.001$, $CI = [0.0261, Inf]$, $d = 0.575$). Oscillating pattern in the predictive performance of Day 8 and Day 9 models on blocks governed by Day 8 and Day 9 statistics indicates that participants successfully recruit different previously learned models for different stimulus statistics (Fig 5E and S8 Fig; note however, that there is high variance across participants in the level of oscillation indicating varying level of success).

In summary, these results demonstrate that the internal model inferred by CT fulfils two critical criteria: the internal model component is general across tasks but is specific to stimulus statistics.

## Evolution of the internal model with increased exposure

Our initial analyses demonstrated that the internal model captured by CT can account for a large component of the variance observed in the responses of participants and also that the ideal observer model can only account for a fraction of this variance. This is expected, since learning the model underlying observations entails that participants need to learn the number of states, the dynamics, and observation distributions, which requires substantial exposure to stimulus statistics. When data is insufficient for an observer to infer the model underlying observations, they can recruit inductive biases that can reflect earlier experiences. The structure of such inductive biases can be very rich. Instead of trying to explore the space of potential forms of inductive biases, we use an Ansatz that is a parsimonious explanation of temporal dependencies, the Markov model. The Markov model only learns immediate dependencies between subsequent observations, which is not in line with the statistics of the applied stimulus sequence but reflects the regularities found in everyday stimuli. In summary, we assume that the gap between the predictive performance of CT and that of the ideal observer can be accounted for by the Markov model. Further, if it constitutes an inductive bias then responses early in the training are governed by Markov model and only gradually wanes. We analyzed the learning curves of individuals through the eight days of training. Our initial analyses were extended with an additional model, the Markov model (Fig 6A). For reference, we also analyzed the trigram model, which can capture essential summary statistics of the stimuli. The predictive performance of the trigram model closely follows that of the ideal observer, indicating that the summary statistics captured by the trigram model is indeed responsible for a substantial part of the statistics reflected by the ideal observer (Fig 6A). The Markov model can capture a significant amount of variance from response times on the first day of exposure ($M = 0.0677$ ranging from 0.00049 to 0.13, one-sided $t(24) = 11.68$, $p < 0.001$, $CI = [0.205, Inf]$, $d = 2.34$), and its performance is not different from that of CT (binomial test on MSE values 0.72; $n = 25$, $p = 0.0433$). Note, that the Markov model is a special case of the model class represented by CT (Fig 3B), therefore indistinguishable predictive performance of the two indicates that the internal models on the first day of training are dominated by a Markov structure.

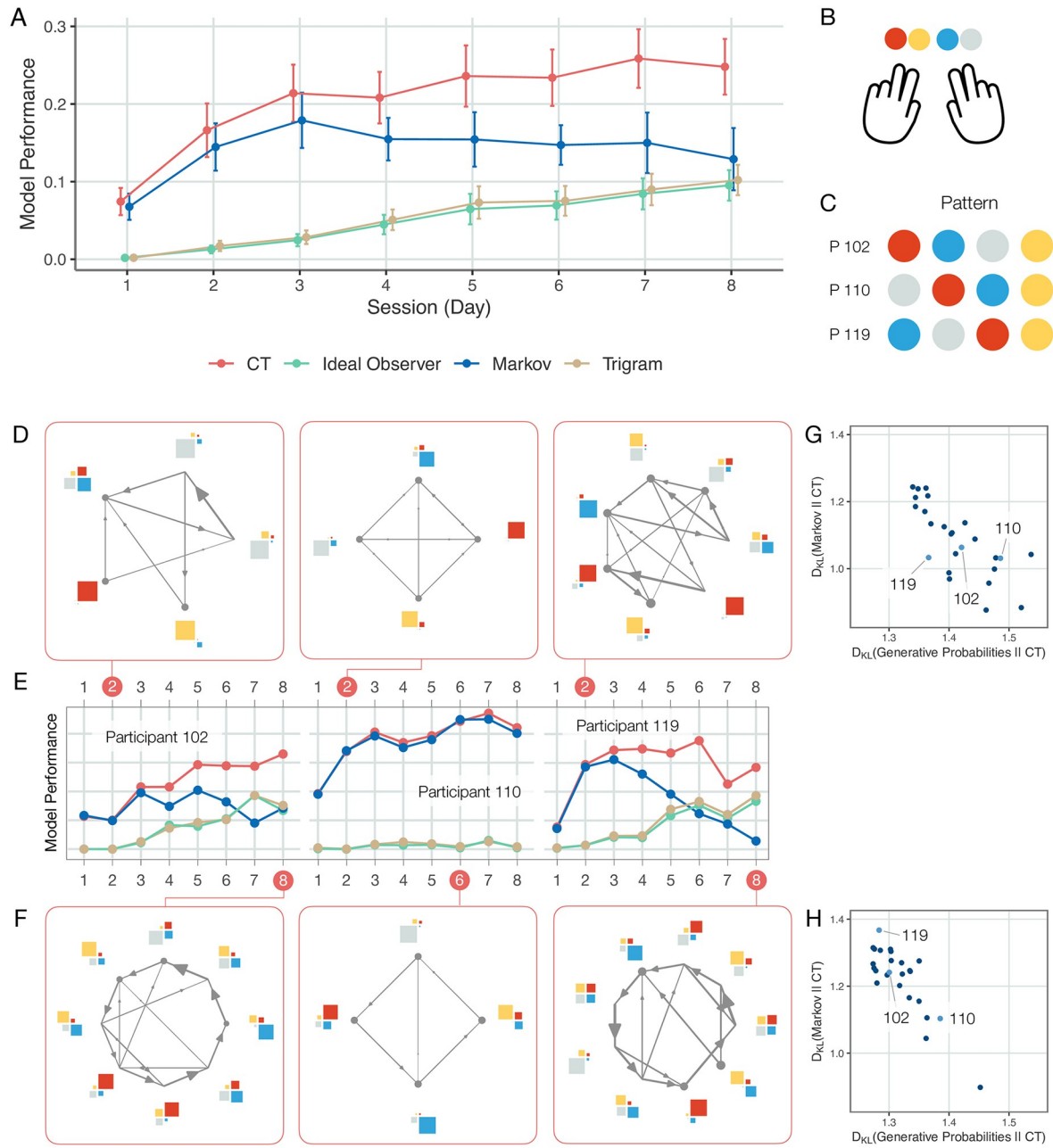

**Fig 6. Evolution of the internal model with increasing training. A** Mean explained variance (*dots*, averaged over participants) in held-out response times in sessions recorded on successive days for the CT (*red*), Markov (*blue*), ideal observer (*green*) and trigram (*yellow*) models. Error bars denote 2 standard error of the group mean. Error bars show 2 s.e.m. **B** Color coding of response buttons used in this figure. **C** Color coding of sequence showed to participants. **D-F** Learning in individual participants (*left*, *middle*, and *right* panels corresponding to different participants: 102, 110, and 119, respectively). **E** Learning curves of CT, ideal observer, Markov, and trigram models. Internal models shown on **D** & **F** panels (corresponding to Days indicated by *red disks* on panel **E**, respectively) are samples from the posterior of possible internal models inferred by CT. CT predictive performance is calculated by averaging over the predictive performances of 60 samples. Participant 102 finds a partially accurate model by Day 2 (**D**) and a model close to the true model by Day 8 (**F**). Participant 110 retains a Markov model throughout the eight days of exposure. Prediction of their behaviour by the Markov model gradually improves while the predictive performance of the ideal observer model is floored, indicating that no higher-order statistical structure was learned. **G** & **H** Mismatch between subjective probabilities of upcoming stimuli derived from CT and alternative models: the ideal observer model (generative probabilities, *horizontal axis*); and the Markov model (*vertical axis*). KL-divergences of the predictive probabilities are shown for individual participants (*dots*) on Day 2 (**G**) and Day 8 (**H**). KL-divergence is zero at perfect match and grows with increasing mismatch.

CT offers a tool to investigate the specific structure of the model governing the behaviour of individuals at different points during learning while exposed to the same sequential statistical structure (Fig 6B–6D and 6F). We computed learning curves of individuals (Fig 6E and S9 Fig) and analysed the internal model structure at different points during training by taking posterior samples from the CT model. Early during training where the predictive performance of the Markov model is close to that of CT, the inferred iHMM indeed tends to have a structure close to that of the Markov model (see also Fig 3B), which is characterized by a strong correspondence between observations and latent states (Fig 6D). Later in the experiment, however, the performance of CT deviates from that of the Markov model for most of the participants (Fig 6E and S9 Fig) and the model underlying the responses reflects a more complex structure (Fig 6F). Note that the monotonic improvement of CT performance can hide a richer learning dynamics: several participants have strong nonlinearities in their learning as initial improvements correspond to a stronger reliance on a Markov-like structure, which is later abandoned for a more ideal observer-like structure (Fig 6F and S9 Fig). Importantly, learning curves and internal models corresponding to different parts of the learning curve reveal qualitative differences between participants. There are participants where improved predictability of response times does not correspond to adopting a model structure that reflects the real stimulus statistics, but the model underlying response times still closely resembles a Markov model (participant 110, Fig 6F, see also Fig 3B). In the meantime, subjects can be identified where the contribution of the Markov model to the internal model declines to almost zero and their internal model seems to faithfully reflect the characteristics of the ideal observer model (subject 119, Fig 6F, note the alternating states with uniformly distributed observations and those with close to certain prediction of observations).

An objective measure of the match between the subjective probabilities of upcoming stimuli and the ground truth probabilities can be obtained by calculating the KL-divergence between the two, a measure commonly used to compare probability distributions. An alternative argument can also be made for using KL-divergence deduced from the LATER model (see S2 Appendix). We computed the KL-divergence between the ground truth probabilities of the task and those of the inferred CT model (Fig 6G) as well as the inferred Markov model and the CT model (Fig 6H), which quantifies the influence of the Markov model on the internal model. The analysis confirms that some participants move away from a Markov model and towards the ground truth probabilities (e.g. participants 102 and 119) while others maintain a model closer to a Markov model throughout the experiment (e.g. participant 110).

## Trade-off between ideal observer and Markov model contributions

The Markov model was shown to be present in multiple days of exposure to the stimulus sequence, and even the internal models of individuals inferred by CT indicated that a Markovian structure largely determines the behavior of individuals early in the training. We assessed the relative contributions of the Markov and ideal observer models by calculating the number of individuals for whom the Markov or the ideal observer model showed higher predictive performance. The Markov model could be identified for all of the participants (Fig 7A), albeit its strength to predict responses varied across participants (S8 Fig). This, along with the observation that the contribution of the Markov model could decline and even diminish for several participants, raised the possibility that the Markov model could constitute the inductive bias participants were relying on.

To investigate this hypothesis, we first tested if the predictive performance of CT can be understood as a combination of the performances of the Markov and ideal observer models. For this, we capitalize on the insight that the Markov and the ideal observer models capture

orthogonal aspects of the response statistics. The Markov model can only account for first order transitions across observed stimuli. The ideal observer model is sensitive to both first-order and second-order transitions but since parameters of the ideal observer model are determined by the stimulus statistics, which lack first-order dependencies the structure that this model actually captures is only sensitive to second-order transitions (Fig 3). As a consequence, the variances in response times explained by these two models are additive.

We used the additivity of Markov and ideal observer variances to assess how well the performance of CT can be predicted by combining the predictions of the ideal observer and Markov models. We contrasted the normalized CT performance, the difference of the variance explained by CT and the Markov model, with the variance explained by the ideal observer model on a participant by participant basis (Fig 7B). We chose to contrast the normalized CT with ideal observer instead of contrasting CT with the sum of the ideal observer and Markov models because this measure emphasizes the contribution of stimulus-statistics to the internal model maintained by participants. We found strong correlation between the two measures ($r(23) = 0.88$, $p < 0.001$), indicating that CT performance can be largely explained by a combination of the Markov and ideal observer models. This strong correlation was consistently present on all recording days ($r(198) = 0.923$, $p < 0.001$, S10 Fig). S9 Fig also reveals that advantage of CT predictive performance over the Markov model only starts to grow as the ideal observer model can be identified in the responses of participants.

A closer inspection of the response time data can provide exquisite insight into how the inductive bias and evidence-based models are combined to determine responses. In particular, we wanted to assess if trial-by-trial CT predictions can be broken down into the individual contributions of the Markov and ideal observer models. We modelled the response time predicted by CT as a linear combination of the predictions obtained by the Markov and ideal observer models. Response times in all trials of a particular participant for any given day were fitted with three parameters: the weight of the contributing models and an offset. Combined response time predictions showed high level of correlation with the predictions obtained from CT: on any given day the across-participant average correlation was close or above 0.8 (Fig 7C). Thus, despite the changing contributions of Markov and ideal observer models across days (Fig 7A), the two models could consistently explain a very large portion of the statistics captured by CT.

We investigated if the statistical structure captured by CT goes beyond that captured by the Markov and ideal observer models. The normalized CT showed a small but significant advantage over the ideal observer model on day eight of the experiment (one-sided $t(24) = 3.646$, $p < 0.001$, $CI = [0.0126, Inf]$, $d = 0.729$, Fig 7D). Therefore we sought to understand if the marginal advantage of the normalized CT predictive performance reflected relevant stimulus statistics that could be captured by CT but not by the Markov or ideal observer models. We analyzed response times to the third element of three-stimulus sequences which the trigram model is unable to distinguish. In one of the analysed conditions, the first and third elements were pattern elements and we compared these to a condition where the first and third elements are random elements but the actual observations were the same. Since only the latent state differed between the two conditions, these cannot be distinguished by the trigram model. Higher order learning, characterised by response time difference between the two conditions, was highly correlated with the higher-order statistical learning predictions of CT both early in the training (Fig 7E, $r(22) = 0.756$, $p < 0.001$) and on the last day of training (Fig 7E, $r(23) = 0.603$, $p = 0.0014$). Interestingly, early in the training most of those participants whose higher-order statistical learning measure was significantly different from zero had negative score (Fig 7E, orange dots), a counter-intuitive finding termed inverse learning [41, 42]. In contrast, higher order statistical learning could not be predicted by the ideal observer (Fig 7E, $r(22) =$

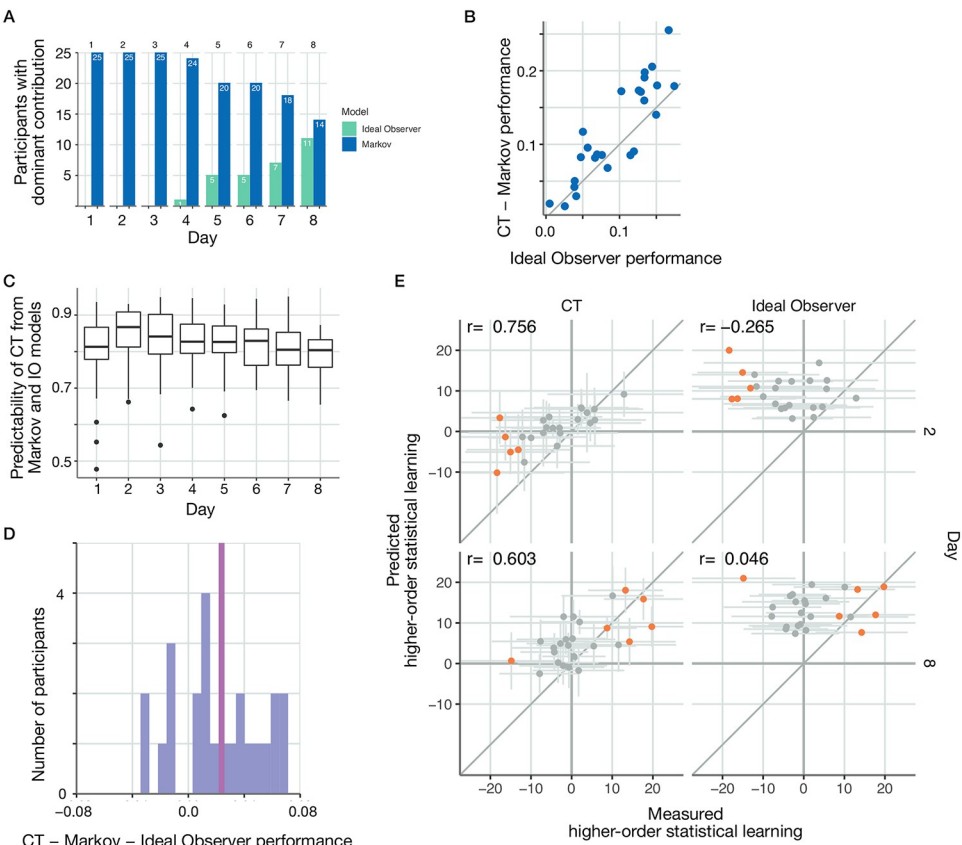

**Fig 7. The internal model captured by CT can be reliably broken down into the independent contribution of an inductive bias and the ideal observer model. A** Day-by-day comparison of the number of participants for whom the predictive performance of Markov (*blue*) or ideal observer (*green*) models was higher. **B** Subject-by-subject comparison (*dots* represent individual subjects) of ideal observer model performance and normalized CT performance (the margin by which CT outperforms the Markov model) on Day 8. Dots close to the identity line (*grey line*) indicate cases where CT performance can be reliably accounted for by contributions from the two simpler models. Normalized CT performance closely follows the performance of the ideal observer model, and deviations tend to indicate slightly better normalized CT performance. **C** Performance of a linear model predicting CT model predictions on a trial-by-trial basis from a Markov and ideal observer model predictions on different days of the training. Thick mid-line indicates $R^2$ of the trial-by-trial fit of the linear combination to CT performance averaged across participants. Boxes show 25th and 75th percentile of the distribution. Upper whiskers show largest value within 1.5 from 75th percentile. Similarly for lower whisker. Dots are data points outside the whiskers. **D** Histogram of the advantage of normalized CT performance over the ideal observer model. Red line marks the mean of the histogram. **E** Higher-order statistical learning in CT (*left panels*) and ideal observer model (*right panels*) on Day 2 (*top panels*) and Day 8 (*bottom panels*) of the experiment. Dots show individual participants. Orange dots represent participants with higher-order learning score significantly deviating from zero. CT can capture both negative deviations (Day 2) and positive deviations (Day 8) in this test and displays significant correlations across participants on both days between the predicted and measured higher-order statistical learning, indicating that subtle and nontrivial statistics of the internal model is represented in CT.

−0.265, *p* = 0.21 and *r*(23) = 0.0457, *p* = 0.828 on Days 2 and 8 of training, respectively). On Day 2, some participants show a significant distinction between these Pattern and Random trials in the reverse direction: responding to Random trials faster than for Pattern trials. The ideal observer model cannot capture this feature of the data whereas internal models inferred using CT can do so.

In summary, while the ideal model cannot account for the full statistical structure captured by CT but together with the Markov model the two models explain the majority of the CT's

internal model such that the relative contributions of the two models shifts towards the ideal observer model during learning.

## Discussion

In this paper we built on the idea of Cognitive Tomography [27], which aims to reverse engineer internal models from behavioural responses, and extended it to infer high-dimensional dynamical internal models from response time data alone. Key to our approach was the combination of non-parametric Bayesian methods that allow discovering flexible latent variable models, with probabilistic programming, which allows efficient inference in probabilistic models. The proposed model has a number of appealing properties for studying how acquired knowledge about a specific domain affects momentary decisions of biological agents: 1, We used iHMM, a dynamical probabilistic model that can naturally accommodate rich inter-trial dependencies, characteristic of an array of everyday tasks; 2, iHMM is capable of capturing arbitrarily complex statistical structure but not increasing the complexity of the model more than necessary [43, 44]; 3, Response times can be predicted on a trial-by-trial basis; 4, Complex individualised internal models could be inferred, which allowed inference of individual learning curves. Using this tool, we could track the deviation of the learned internal model from the ideal observer model. We identified transient structures that were nurtured temporarily only to be abandoned later during training. The deviation could be consistently explained by the contribution of a simpler model, a so-called Markov model, that learns the immediate temporal dependencies between observations but ignores latent variables. Initial dominance of the internal model by the Markov model indicated that the Markov model constitutes an inductive bias that humans fall back to when experience is limited. Indeed, during learning the contribution of Markov model decreased on an individual-by-individual basis, which coincided with a gradual decrease in the deviation between the inferred internal model and the ideal observer model.

Learning in general is an ill-defined, under-determined problem. Learning requires inductive biases formulated as priors in Bayesian models to efficiently support the acquisition of models underlying the data [45, 46]. The nature of such inductive biases is a fundamental question which concerns both cognitive science and neuroscience, even machine learning [45, 47, 48]. These inductive biases determine what we can learn effectively. Inductive biases can effectively support learning if these represent priors, which reflect the statistics of the environment. Indeed, Markovian dynamics can be a good approximation of the dynamics of the natural environment therefore can constitute a useful inductive bias. Our analysis demonstrated that participants are remarkably slow to learn the ground truth statistics of the stimuli. Our results also showed that this slow learning dynamics can be accounted for by a strong inductive bias that is consistent across participants. Slow acquisition of the true task statistics might indicate the low a priori probability of the task statistics among the potential hypotheses humans entertain. The spectrum of inductive biases that humans use can be much richer than the Markov model. For instance, after the extended exposure to the ASRT, one can expect that the inductive biases can be updated. It will be an exciting future line of research how we can identify updates in inductive biases, a question related to the broader topic of meta learning, or transfer learning [49].

The model class that we use to infer the internal model has a strong effect on the types of statistics in the data that can be learned effectively. Our proposed model class, iHMM, is appealing because it can accommodate highly complex statistical structures, including the ideal observer model or the Markov model. The flexibility of the model comes at a price that more data is required for inferring the model. This motivated the choice to infer one model

per session and track learning by comparing inferred models across days. Our choice is motivated by the revealed multi-day learning process that seemed to be characteristic of all participants. The approach has a limitation too, that early in the training (and especially on the first day) faster changes in the internal model cannot be captured. Alternative model classes, such as a hierarchical version that can more effectively perform chunking, can be more effective in learning from more limited data and can be used to explore the evolution of the internal model in more detail [50]. Data hunger of model inference can be further curbed by learning a constrained version of parameters, such as the transition matrix) but at the expense of hindering potential individual differences. The proposed framework of cognitive tomography naturally accommodates such alternative model classes and we expect further insights into the way inductive biases are used during learning.

The presented model builds on the original CT analysis performed on faces [27] but differs in a number of fundamental ways. We sought to infer the evolution of the internal model for statistics new to participants. In contrast to the earlier formulation using a 2-dimensional latent space and a static model, here the inference of a dynamical and potentially high-dimensional model yields a much richer insight into the working of the internal model acquired by humans. Using a structured internal model allows the direct testing of the model against alternatives, thus providing opportunities to reveal the computational constraints that might limit learning and inference in individuals. A well-structured internal model can be used to make arbitrary domain-related inferences within the same model. Based on this, we can decompose the complex inference problem into separately meaningful sub-parts which can be reused in tangential inference problems to serve multiple goals. Our experimental design permitted some exploration of such across-task generalization capabilities, but suitably updated alternative designs could provide a more exhaustive test of across-task generalization. By showing that the same variables can be used for multiple tasks, it is reasonable to look for signatures of these quantities in neural representations. A possible alternative formalization of this problem could be using Partially Observed Markov Decision Process (POMDP) [51], where internal reward structure and the subjective belief of the effect of the participant's actions are jointly inferred with the internal model. However, in our experiment, the action model has a simple structure and hence the problem simplifies to a probabilistic sequence learning problem. Instead, here we focus on inferring rich internal model structures as well as having an approximate Bayesian estimate instead of point estimates as in [51]. Still, the ability of POMDP to model how decisions of the agent affect the state of the state of the sequence can become useful for investigating the potential inductive bias that actions actually influence states.

Our model produces moment by moment regressors for (potentially unobserved) variables that are cognitively relevant. Earlier work considered neural correlates of hidden state representations in the orbitofrontal cortex of humans [52] but the internal model was not inferred, rather assumed to be fully known. CT provides an opportunity to design regressors for individualised and potentially changing internal models. In particular, the model differentiates between objective and subjective uncertainties, characteristics relevant to relate cognitive variables to neural responses [53–56]. The former is akin to a dice-throw, uncertainty about future outcomes which may not be reduced with more information. The latter is uncertainty arising from ambiguity and lack of information about the true current state of the environment. We showed that uncertainties exhibited by a trained individual's internal model show similar patterns in these characteristics as the ideal observer model, which promises that uncertainties inferred at intermediate stages of learning are meaningful.

Recently, major efforts have been devoted to learning structured models of complex data both in machine learning and in cognitive science [9, 57–59]. These problems are as diverse as learning to learn [57, 60], causal learning [61], learning flexible representational structures

[58], visual learning [62]. When applied to human data to reverse engineer the internal models harnessed by humans, past efforts fall into two major categories. 1, Complex (multidimensional) models are inferred from data and fitted to across-participant averaged data [9, 63, 64], ignoring individual differences. 2, Simple (low dimensional) models are used to predict performance on a participant-by-participant manner, thus resulting in subjective internal models [29, 65]. In particular in a simple two-latent variable dynamical probabilistic model individualised priors have been identified [11]. In this setting binary decisions were sufficient as an 'expert model' was assumed and assessment of prior comprised of inferring a single parameter, which defined the width of a one-dimensional hypothesis space. Findings of this study gave insights into how individuals differ in their capacity to adapt to new situations. Recently, a notable approach has been presented, which aims at characterizing individual strategies in a setting where the complexity of the state space is relatively large [66]. In this study, the rules of the game (equivalent of the statistics of stimuli in our case) and the relevant features (equivalent to the latent variables in our case) were assumed to be known by the participants. However, being a two-player task there was uncertainty about the strategy of the opponent and the limitations in the computational complexity of the inference was investigated. This aspect is orthogonal to the aspects investigated here and therefore highlight additional appeal of analysing behavior in complex settings. The contribution of the current paper is twofold: 1, We exploit recent advances in machine learning to solve the reverse-engineering problem in a setting where complex internal models with high-dimensional latent spaces are required; 2, We contribute to the problem of identifying structured inductive biases by enabling direct access to the internal model learned by individuals and by dissecting the contributions of evidence and inductive bias.

A widely studied approach to link response time to quantities relevant to task execution is the drift diffusion model, DDM [67]. In its most basic form evidence is stochastically accumulated as time passes such that the rate of accumulation is proportional to the information gained by extended exposure to a stimuli, until evidence reaches a bound where decision is made. Through a compact set of parameters DDM can explain a range of behavioural phenomena, such as decisions under variations in perceptual variables, adaptation to the volatility of the environment, attentional effects on decision making, the contribution of memory processes to decision making, decision making under time pressure [68–72], and neuronal activity was also shown to display strong correlation with model variables [73, 74]. Both LATER and DDM have the potential to incorporate variables relevant to make decisions under uncertainty and the marginal distributions predicted by the two models are comparable. Our choice to use the LATER model was motivated by two major factors. First, LATER is formulated with explicit representation of subjective predictive probability by mapping it onto a single variable of the model. This setting promises that subjective probability can be independently inferred from available data and the internal model influences a single parameter of the model. As a consequence, subjective probability is formally disentangled from other parameters affecting response times and associated uncertainty can be captured with Bayesian inference. In case of distributing the effect of subjective probability among more than one parameters (starting point, slope, variance) the joint inference of subjective probability with other parameters affecting response times results in correlated distributions. Consequently, maximum likelihood inference, or any other point estimations, the preferred method to fit DDM, will have large uncertainty over the true parameters due to interactions between other variables. Furthermore, this uncertainty remains unnoticed as there is usually no estimation of this uncertainty, only point estimates. Second, trials are usually sorted based on the design of the experiment into more and less predictable trials (with notable exceptions like [29]). This leads to a misalignment between the true subjective probabilities of a naive participant and the

experimenter's assumptions. Assuming full knowledge of the task and therefore assuming an impeccable internal model in more complex tasks, however, implies that potential variance in the acquired internal models across subjects will be captured in variances in parameters characteristic of the response time model rather than those of the internal model. DDM is considered to be an algorithmic-level model [75] of choices [76], which is indeed useful for linking choice behaviour to neuronal responses [77]. The appeal of the Bayesian description offered by the normative framework used here is that it can accommodate a flexible class of internal models, without the need to adopt algorithmic constraints. Similar algorithmic-level models of behaviour that is based on the flexible and complex internal models yielded by Cognitive Tomography are not available and will be the subject of future research.

In summary, we presented and validated a tool that could flexibly infer complex, dynamical, individualised internal models from simple behavioral data. We demonstrated that various levels of discrepancy existed between the ideal observer model and the internal model maintained by individuals. We used this discrepancy to identify an inductive bias with a structure that was consistent across participants. This approach promises that altered contribution of inductive biases or learning can be identified in affected populations at the individual level. An additional promise provided by the presented approach is the separation of the internal model from the behavioral models: the Cognitive Tomography framework can naturally integrate diverse behavioral data into a single model, thus by using multiple modalities ensures faster and more accurate inference of the internal model.

## Methods

### Ethics statement

All participants provided written informed consent before enrollment and received course credits for taking part in the experiment. The study was approved by the United Ethical Review Committee for Research in Psychology (EPKEB) in Hungary (Approval number: 30/2012) and by the research ethics committee of Eötvös Loránd University, Budapest, Hungary. The study was conducted in accordance with the Declaration of Helsinki.

### Experiment

**Participants.**   Twenty-five individuals (22 females and 3 males) aged between 18 and 22 ($M_{Age}$ = 20.4 years, $SD_{Age}$ = 1.0 years) took part in the experiment (we recruited 32 participants, but only 26 completed the experiment; we omitted one further participant because of a system error which resulted in partial loss of their experiment data). They were university students ($M_{\text{Years of education}}$ = 13.3 years, $SD_{\text{education}}$ = 1.0 years) from Budapest, Hungary. None of the participants reported history of developmental, psychiatric, neurological or sleep disorders, and they had normal or corrected-to-normal vision. They performed in the normal range on standard neuropsychological tests of short-term and working memory (Digit span task: $M$ = 6.48, $SD$ = 1.15, Counting span task: $M$ = 3.76, $SD$ = 0.99) [78]. Before the assessment, all participants gave signed informed consent and received course credit for participation.

**Tasks.**   Alternating Serial Reaction Time (ASRT) Task Learning was measured by the ASRT task [35, 79]. In this task, a stimulus (a dog's head) appeared in one of four horizontally arranged empty circles on the screen and participants were asked to press the corresponding button as quickly and accurately as they could when the stimulus occurred. The computer was equipped with a keyboard with four heightened keys (Z, C, B, M on a QWERTY keyboard), each corresponding to a circle in a horizontal arrangement. Participants were asked to respond to the stimuli using their middle- and index fingers bimanually. The stimulus remained on the screen until the participant pressed the correct button. The next stimulus appeared after a 120

ms response-to-stimulus-interval (RSI). The task was presented in blocks of 85 stimuli: unbe-knownst to the participants, after the first five warm-up trials consisting of random stimuli, an 8-element alternating sequence was presented ten times (e.g., 2r4r3r1r, where each number represents one of the four circles on the screen and r represents a randomly selected circle out of the four possible ones). The sequence started at the same phase in each block.

**Procedure.** There were ten sessions in the experiment, with one-week delay between the consecutive sessions. Participants performed the ASRT task with the same sequence in the first eight sessions, then an interfering sequence was introduced in Session 9, and both (original and interfering) sequences were tested in Session 10 (see S1 Fig). Participants were not given any information about the regularity that was embedded in the task in any of the sessions [79]. They were informed that the main aim of the study was to test how extended practice affected performance on a simple reaction time task. Therefore, we emphasized performing the task as accurately and as fast as they could. Between blocks, the participants received feedback about their average accuracy and reaction time presented on the screen, and then they had a rest period of between 10 and 20 s before starting the next block. On Days 1–9, the ASRT consisted of 25 blocks. One block took about 1–1.5 min, therefore the task took approximately 30 min. For each participant, one of the six unique permutations of the four possible ASRT sequence stimuli was selected in a pseudo-random manner [35, 79, 80]. The ASRT task was performed with the same pattern sequence in Sessions 1–8. In Session 9, the ASRT was performed with a new interfering pattern sequence. In Session 10, participants performed 20 blocks of the ASRT task switching between the pattern sequences of Sessions 1–8 and Session 9 every five blocks. In Session 10, the task took approximately 24 min. After performing the ASRT task in Session 10, we tested the amount of explicit knowledge the participants acquired about the task with a short questionnaire. This short questionnaire [79, 81] included two questions: "Have you noticed anything special regarding the task?" and "Have you noticed some regularity in the sequence of stimuli?". The participants did not discover the true probabilistic sequence structure.

## Modelling background

**Models for sequential prediction.** The experimental stimuli form a sequence of discrete observations in discrete time, $\{Y_t\}_{t=1}^{T}$. The task is therefore to predict the upcoming stimulus conditioned on the history of observations:

$$P(Y_{T+1}|Y_1, Y_2, \ldots, Y_T) \tag{1}$$

In practical terms, learning a model for this temporal prediction task requires imposing a structure over these conditional distributions. Without structural assumptions, there is no sta-tistical dependence among different histories, that is, there is no generalisation from history to future observations.

In the following section we introduce a computational model, the Hidden Markov Model, which can provide a general language for solutions of this problem. It can express arbitrarily complex models given sufficiently large amounts of data. In order to remain as general as pos-sible, we will consider a model space (infinite Hidden Markov Models as in [33]) which can model all the possible distributions in Eq 1. Moreover, we would like to achieve this while being able to express inductive biases in this language which are useful for constraining the possible models in the limited data case.

**Hidden Markov model.** Formally, a Hidden Markov Model comprises of a sequence of hidden states $\{S_t\}_{t=1}^{\infty}$ and a sequence of observations $\{Y_t\}_{t=1}^{\infty}$. In this work we take both the latent states and the observations to be discrete, that is $S_t, Y_t \in \mathbb{N}$. The sequence of hidden

(latent) states constitute a discrete Markov-chain with transition probabilities $\pi_{ij} = P(S_{t+1} = j|S_t = i)$. In a Markov-chain, the sequence element $S_t$ is conditionally independent of the history conditioned on the previous state and the transition probabilities:

$$S_t \perp (S_1, S_2, \ldots, S_{t-2}) \mid S_{t-1}, \pi$$

At (discrete) time $t$, observation $Y_t$ is governed by the latent state $S_t$. The observations are generated independently and identically, conditioned on the (latent) state:

$$P(Y_t = y \mid S_t = s_t) = \phi_{s_t,y} \quad \text{and} \quad Y_t \perp (Y_1, Y_2, \ldots, Y_{t-1}, S_1, S_2, \ldots, S_{t-1}) \mid S_t, \phi$$

Importantly, since the latent state can incorporate arbitrary information (identical observations at different time-points can correspond to different states), assuming arbitrarily many latent states, we get a completely general solution for the prediction problem in Eq 1. With an adequate prior (e.g. the Hierarchical Dirichlet Process in [82] we can learn such structures efficiently [33]). In practical terms the length of the observation sequence limits the number of possible latent states but it is limited by the diminishing posterior probability of high latent state models.

## Cognitive tomography

We construct a model of behaviour which consists of two parts:

1. An internal model maintained by the participant, which formalizes how latent states assumed to underlie observations evolve and how these states are linked to observations.

2. A model relating the prediction of participants' internal model to their responses (response time model).

**Doubly Bayesian model.** Due to the uncertainty of the participants about the true model and actual state in the stimulus sequence and to the uncertainty of the experimenter about the model maintained by participants and about the actual state of this internal model, the problem can be described as doubly Bayesian. We do Bayesian inference over an internal representation of individuals who themselves do Bayesian inference. Elements of the experimenter's model are introduced in following sections.

Prediction of response times can be described by the following algorithm:

1. We take posterior samples from the behavioural model which consists of parameters of the internal model and the response time model conditioned on data from ten consecutive blocks of trials (see explanation for ten below), where:

    (a). all stimuli, and

    (b). response times (with incorrect trials', first five random trials' response times, and response times smaller than 180 msec in each block removed). According to the original formulation by [34], fast response times come from an alternative distribution. We cut off the fast response times (as in [83]) at the fixed 180 msec value. However, we did not fit the cut-off time parameter. Incorrect trials constitute 11% of trials overall while trials below the 180msec threshold constitute 2.2% of trials overall and 5.3% on Day 8. are included.

2. For each of the posterior model samples we compute predicted response times by:

    (a). filtering the belief over the latent state over the entire sequence

(b).  produce subjective probabilities for each trial

(c).  produce response time prediction (MAP estimate conditioned on the subjective probability and the response time parameters of the model sample)

Then we marginalize (i.e. average) over the response time predictions of model samples.

3. We evaluate model performance by computing the $R^2$ explained variance measure of the predicted response times on the response times of the test dataset. In any given session we train the model on one set of blocks and predict response times on a distinct set of test blocks. During optimizing our model and algorithm, we concluded that using ten consecutive blocks for training provides the best results for the CT model. We also found that using ten blocks for the test set decreases variance of the $R^2$ estimator sufficiently to have individualised learning trajectories.

Note: since actual beliefs depend on past beliefs, one can think of the belief sequence as the path of a light-ray in a large dimensional fog (representing the state uncertainty). During inference, we have a noisy measurement of the light-ray in different points of time and we would like to reconstruct the best explanation of the observation sequence (response times) in terms of a hidden path. As for prediction, the model produces response time predictions for the entire stimulus sequence with no further feedback of response times (i.e. estimated internal beliefs are not updated based on what response time the participant produced on given trials).

**Infinite Hidden Markov model.** The infinite Hidden Markov Model is a non-parametric extension of the Hidden Markov Model, assuming countably infinitely many states. There is a hierarchical prior imposed over the state transition matrix and the so-called emission distributions relating the latent (hidden) states to observations (S2 Fig).

The hierarchical prior we used is exactly the one defined in [33]. We extended their implementation of their model to a doubly Bayesian behavioural model including the response time.

A participant is assumed to learn a probabilistic model of the sequence which is formalized as an infinite Hidden Markov Model. At (discrete) time $t$, observation $Y_t$ is governed by a latent (not directly observable) state $S_t$. The states $\{S_t\}_{t=1,2,\ldots}$ constitute a Markov-chain, which means the following:

$$p(S_t|S_1, S_2, \ldots, S_{t-1}) = p(S_t|S_{t-1}) \tag{4}$$

That is, the state $S_{t-1}$ holds all information about past regarding the possible evolution of system. In other terms, conditioning on state $S_{t-1}$ renders $S_t$ and all previous states $S_1, S_2, \ldots, S_{t-2}$ statistically independent.

The observation $Y_t$ at time $t$ is independent of all other observations, conditioned on the latent state $S_t$ (and the model parameters). That is, once the state of the system is decided, the actual previous observations are independent of $Y_t$.

The parameters governing the state transitions are aggregated in the parameter matrix $\pi$:

$$\pi_{i,j} = p(S_t = j|S_{t-1} = i) \; \forall t$$

The observation distributions are given by the parameter matrix $\phi$:

$$\phi_{i,k} = p(Y_t = k|S_t = i)$$

At any given time during the task, we assume the participant had estimated the parameters $\pi$ and $\phi$ and uses these (point estimates) to do exact filtering over the sequence of observations. That is, in each trial they use the evidence provided by the current stimulus to update their

belief over the latent state of the sequence. When doing computations with the participant's internal model, we hold the internal model fixed within shorter time-scales of the task (e.g. one session). The participant represents their belief about the current latent state of the system by a posterior distribution, updated by each incoming observation, while always conditioning on their current estimates $\hat{\pi}$ and $\hat{\phi}$ of $\pi$ and $\phi$ respectively. We denote this posterior distribution over latent states at time $t$ by $\hat{s}_t$ (note this is not a point estimate of the state but rather a vector of probabilities where $(\hat{s}_t)_i = p(s_t = i)$.

$$
\begin{aligned}
\hat{s}_t := p(s_t|y_1, y_2, \ldots, y_t) \quad &\propto p(y_t|s_t)p(s_t|y_1, y_2, \ldots, y_{t-1}) \\
&= \sum_{s_{t-1}} \hat{\phi}_{s_t,y_t} p(s_t|s_{t-1}) p(s_{t-1}|y_1, \ldots, y_{t-1}) \\
&= \sum_{s_{t-1}} \hat{\phi}_{s_t,y_t} \hat{\pi}_{s_{t-1},s_t} \hat{s}_{t-1}
\end{aligned}
$$

For predicting the latent state based on previous states and the observation (termed filtering), stimuli of all trials (including initial random trials at the beginning of each block and stimuli in trials where participant hit the wrong key initially) are used. That is, even if incorrect response times are not used when doing inference over the participant's internal model, the participant is assumed to update their internal beliefs based on the stimulus shown.

Prediction of the next stimulus is computed by marginalizing over the latent state posterior distribution:

$$
\begin{aligned}
p(y_{t+1}|y_1, y_2, \ldots, y_t) \quad &= \\
&= \sum_{s_{t+1}} p(y_{t+1}|s_{t+1}) p(s_{t+1}|y_1, y_2, \ldots, y_t) \\
&= \sum_{s_{t+1}} \hat{\phi}_{s_{t+1}y_{t+1}} p(s_{t+1}|y_1, y_2, \ldots, y_t) \\
&= \sum_{s_t,s_{t+1}} \hat{\phi}_{s_{t+1}y_{t+1}} p(s_{t+1}|s_t) p(s_t|y_1, y_2, \ldots, y_t) \\
&= \sum_{s_t,s_{t+1}} \hat{\phi}_{s_{t+1}y_{t+1}} \hat{\pi}_{s_t,s_{t+1}} \hat{s}_t
\end{aligned}
$$

Throughout the execution of the task, the internal model of the participants is continually updating. We do not directly model the computation of the participants that estimates the current $\pi$ and $\phi$ parameters. That is, within a given train or test dataset (10 consecutive blocks) we hold $\pi$ and $\phi$ fixed. We do allow, however, for these estimates of $\pi$ and $\phi$ to change between sessions. For a summary of when each parameter is allowed to change see Table 1.

**Table 1. Summary of when model parameters are allowed to change.**

| Variable | Notation | Within train/test | Within session, between train-test | Between sessions | Between participants |
|---|---|---|---|---|---|
| State transition distribution | $\hat{\pi}$ | No | No | Yes | Yes |
| Observation distribution | $\hat{\phi}$ | No | No | Yes | Yes |
| State belief | $\hat{s}_t$ | Yes | Yes | Yes | Yes |
| Response time parameters | $\tau_0, \mu, \sigma$ | No | No | No | No |
| Prior of observation distribution | $H$ | No | No | No | No |
| Hierarchical prior over state transitions | $\alpha, \gamma$ | No | No | No | No |

**Table 2. Parameter priors.** Values of the hierarchical prior over state transitions taken from [33].

| Variable | Prior |
|---|---|
| State transition distribution | $\hat{\pi}_i \sim \text{Dirichlet}(\alpha_0/K, \ldots, \alpha_0/K, \alpha_0/K \cdot \epsilon)$ |
| Observation distribution | $\hat{\phi} \sim \text{Dirichlet}(0.8, 0.8, 0.8, 0.8)$ |
| Response time parameters | $\tau_0 \sim \Gamma(1, 10)$ |
|  | $\mu \sim \Gamma(1, 0.1)$ |
|  | $\sigma \sim \Gamma(1, 0.01)$ |
| Hierarchical prior over state transitions | $\alpha = 1.3$ |
|  | $\gamma = 3.8$ |

We do Approximate Bayesian Inference using a custom sampling method that mixes steps of a Hamiltonian Monte Carlo (HMC) and a Gibbs sampler which samples a slicing parameter (see [33]). The priors used in the model are listed in Table 2.

In order to handle the infinitely many possible states, we use a modified version of the slice sampling method described in [33]. In the original beam sampling algorithm, the authors sample the latent state sequence and make use of the slicing variable to constrain the set of used states to a finite set. They sample the latent sequence and the slicing variables in an alternating fashion. In our case we do not sample latent state sequences, instead, we have to estimate the subjective belief sequence over the latent states. In this latter case, the posterior belief is infinite dimensional and we use slicing to approximate this infinite-dimensional computation with a finite one. At each sampling step, we only look at the latent state belief distribution's $1 - \epsilon$ support where $\epsilon$ is sampled from Uniform(0.02, 0.2).

Four independently and randomly initialised Markov Chains were sampled with 1600 steps of the slice sampling (outer Gibbs-sampling chain) and 30 NUTS steps were taken in between the slice sampling steps each time. Samples from the second half of each chain were used to check if estimates of response time parameter means and confidence intervals were identical. For prediction, the last 60 unique samples were used from each chain because prediction performance saturates at this number of samples.

**Ideal observer model.** We formalise the ideal observer the following way: at any given point of the experiment, the ideal observer entertains an internal dynamical model comprising of two parts: latent dynamics (the transition probabilities between latent states) and an observational model (conditional distributions of observations conditioned on the latent state).

In order to produce predictions for the upcoming observation, conditioning on a fixed model, the ideal observer solves the filtering problem:

$$P(Y_t | Y_1, Y_2, \ldots, Y_{t-1}, \pi, \phi) =$$

$$= \sum_{s_t=1}^{\infty} P(Y_t | S_t = s_t) \cdot P(S_t = s_t | Y_1, Y_2, \ldots, Y_{t-1})$$

$$= \sum_{s_t=1}^{\infty} \phi_{s_t, y_t} \cdot P(S_t = s_t | Y_1, Y_2, \ldots, Y_{t-1})$$

$$= \sum_{s_t=1}^{\infty} \sum_{s_{t-1}=1}^{\infty} \phi_{s_t, y_t} \cdot P(S_t = s_t | S_{t-1} = s_{t-1}) \cdot$$
$$P(S_{t-1} = s_{t-1} | Y_1, Y_2, \ldots, Y_{t-1})$$

$$= \sum_{s_t=1}^{\infty} \sum_{s_{t-1}=1}^{\infty} \phi_{s_t, y_t} \cdot \pi_{s_{t-1}, s_t} \cdot P(S_{t-1} = s_{t-1} | Y_1, Y_2, \ldots, Y_{t-1})$$

The term filtering is used because as we deduced, the relevant quantity is $P(S_{t-1}|Y_1, Y_2, \ldots, Y_{t-1})$ which can be filtered through our observations. We carry on this quantity and can calculate it for the next time-step using our model parameters and the observation $Y_t$.

Importantly, instead of sampling one possible latent trajectory, we have to marginalise over these latent sequences to obtain our prediction for the upcoming stimulus. That is, our prediction is the aggregate of the predictions of many possible latent pasts. We combine the predictions of 'had these been the sequence of causes of my past experiences, I should see this' for all possible hypothesised latent cause sequences.

**Response time model.** In order to connect the predictions of the internal model to measured behaviour, we need to employ a generative model of response times in the form of a conditional probability distribution conditioned on the subjective predicted probability of the upcoming stimulus. To achieve this, we employ the reaction time model of [34], which in its original formulation states that the majority of saccadic response times come from a reciprocal Normal distribution.

Further studies suggest choice response time distribution should have a similar form [84, 85]. However, in other formulations, there is no explicit dependence of the distribution of the RT in a single trial depending on the subjective predicted probability, hence those models are inadequate for our purposes. The generative model for correct response times (LATER model, [34]) is:

$$r_n \sim \text{Normal}(\mu, \sigma)$$

$$RT_n = \frac{\theta_0 - \log(p_n)}{r_n}$$

where $p_n$ is the subjective probability (output of the internal model) corresponding to the actual upcoming stimulus and $\mu$, $\sigma$, $\theta_0$ are the parameters characterising an individual's response time model. These parameters jointly describe the mean and variance of the response times. Note that in our experiment these parameters comprise all idiosyncratic effects at hand, namely the individual's state, their response times' sensitivity to subjective predicted probabilities, the effects of instruction influencing speed-accuracy trade-off. Note, that in order to avoid assigning probability to negative reaction times, we use a truncated Normal distribution.

The response time parameters are jointly inferred along with the internal representations (dynamical model, observation distribution, latent state inference).

**Validation on synthetic datasets.** In order to validate our behavioural model as well as our inference method, we looked at how well we can recover subjective probabilities on a synthetic dataset. We chose to constrain our analysis to the recovery of subjective probabilities instead of the generative model structure due to the unsupervised nature of our method: the objective of inference is to learn the distribution of data (which is in direct relationship with the predictive probability of upcoming stimuli). This is in contrast with more supervised methods where the emerging representations can be gauged by performing tasks that rely on the latent variables. We used the algorithm in [33] on synthetic ASRT data to infer a first set of three different internal models from different levels of exposure. These models represent internal models of different synthetic participants (S3(A) Fig). As a prior predictive check, we show marginal distributions of synthetic response times that approximately match response time distributions of humans (S11 Fig). We take these models as the ground truth for our synthetic experiment. We trained one model on 640, 1280 and 2400 trials of ASRT stimuli. We then generated response times from the generative model with three parameter settings for each of $\tau_0$, $\mu$, and $\sigma$ resulting in a total of $3^3 = 27$ different synthetic response time sequences. The

resulting response time distribution's variance is influenced by all four factors—the subjective probabilities (which depends on the internal model) and the three response time parameters. The standard deviation of the response times is an appropriate measure since this can be also computed for data obtained from human participants. We generated the response times for 10 ASRT blocks (the same number we used for inference on human data). Standard deviation of the resulting response times (*symbol colours* on S3(B) Fig) arise from the interaction of all parameters. Different combinations of the response time parameters resulting in the same standard deviation are marked by identical colours. Then, we used the CT inference method to generate a second set of (posterior) internal model samples. We computed the same model performance measure as for human data (response time prediction performance) and compared it to the prediction performance of that of the original internal model of the participant (S3(B) Fig). Then, since the recovered model matched in this performance to the ground truth internal model of the synthetic participants, we also compared how well the actual subjective probabilities of said synthetic participants can be predicted (S3(C) Fig). The results show that the prediction performance of the subjective probabilities exceeds that of the individual response times. Also, as seen in S3(D) Fig, standard deviations of human participant's response times are within the range for which we validated our model inference method.

## Alternative models

**Markov model.   Internal model of participants**. According to this model, the participants assume that the sequence of observations constitute a Markov-chain. That is, for the sequence of observations $y_t$, we have

$$p(y_t|y_1, y_2, \ldots, y_{t-1}) = p(y_t|y_{t-1}) \quad \forall t$$

The above equation states that the next observation is independent of all previous observations given the previous observation. This is equivalent to saying that all information (besides parameters governing the sequence) about the state of the sequence is included in the previous observation.

**Inference**. We use the same parameter priors for the response time model as for the iHMM model and the prior for transition probabilities $\pi_i \sim \text{Dirichlet}(\alpha_0/K, \alpha_0/K, \alpha_0/K, \alpha_0/K)$, where $K$ is the number of states, in this case 4.

Four independently and randomly initialised Markov Chains were sampled with 1600 steps taken with the NUTS sampler in STAN. Samples from the second half of each chain were used to check if response time parameter estimates' means and confidence intervals were identical. For prediction, the last 60 samples were used from each chain.

**Relation to HMM**. Note that Markov models are a subset of Hidden Markov Models. We can always write a Markov model as an HMM if we have a matching number of observation values and latent state values and each observation is unique to a state.

This is particularly important since for an HMM for which the above condition holds, there is an equivalent Markov chain that describes the exact same sequence structure. This is the reason why we term some of the internal models identified by our iHMM method "Markov-like", since they are closely approximated by an actual Markov model.

**Trigram model.**   The model we describe here is also referred to as 'triplet model' in previous works using the ASRT paradigm. We use the term trigram since it is more commonly used in a sequential prediction modelling context.

**Internal model of participants**. The model, established in prior literature, sorts trials into High probability and Low probability triplets. This is equivalent to assuming that the participant uses a two-back (or trigram) model for prediction, predicting the most-likely stimulus

conditioning on the previous two observations. Due to the ground truth generative model of the task there is no practical dependence on the identity of the immediately preceding stimulus, and only the penultimate stimulus can contribute to making predictions.

**Inference**. The trigram model has no parameters fitted. Predictive performance is evaluated by the $R^2$ measure between the response times and the binary variable (high vs low trials) provided by the trigram model.

## Model comparison

Since not all models considered are Bayesian (i.e. provide an explicit marginal log-likelihood for the response times), we chose to compare models based on explained variance of response times on a test set. Each model produces response time predictions for each trial and each individual separately. When evaluating on a given test set, in order to control for a shift in mean not related to the inherent structure of the response times, we use $R^2$ as our performance metric. That is equivalent to assuming that the actual observed response times come from a linear model with the predicted response time as mean and an additive homoscedastic (equal variance irrespective of predicted response time) normal noise term.

$R^2$ values were calculated separately for each individual's trials.

**Train and Test Datasets**. For the reason described in the above paragraph, for each day (out of 10) of the experiment, out of the 25 blocks each day, we selected blocks 11–20 as a training dataset and blocks 1–10 as test datasets. The main reason for this choice is that on each day in the initial few blocks participants may be engaged in a warm-up phenomenon which fundamentally alters their behaviour in the task. If we use the first 10 blocks as test data, the performance metric may be influenced by 10–30% depending on how many blocks include altered behaviour. However, if we used this part as training data, the whole internal model inference would shift fundamentally, since our inference algorithm assumes a fixed model the entirety of the 10 blocks.

During model inference (train dataset) and performance evaluation (test dataset) the first five random trials and all incorrect response trials' response times are not considered.

## Statistical methods

Normality was not checked prior to t-test comparisons. All reported correlations were computed using Pearson's correlation. T-tests are paired sample tests whenever there is a within-subject comparison. All binomial tests are one-sided. For effect sizes we calculated Cohen's d using the lsr R package.

## Error prediction

In the error prediction task we analyzed trials in which participants did not press the button corresponding to the actual stimulus and instead pressed a wrong button. The analysis assesses two quantities: the subjective probability of the correct buttons relative to that of the other buttons, and the subjective probability of the erroneously pressed button relative to those of other buttons. Just as with response time prediction, the model outputs (for each posterior model sample) a subjective probability estimate for each one of the four possible stimuli for each trial. Then, we take the mean over these probability estimates over the last 60 unique samples of each chain. We decided on using 60 samples since model performances saturate at this number. In Fig 5 we compute the rank among the four probability estimates of the stimulus and the choice in correct and incorrect trials. Then, based on the subjective probability estimates of the actual occurring stimulus, we plot the receiver operating characteristic curve for predicting whether a given trial will result in an error. This is done by moving a threshold value from 0 to 1 and predicting a correct trial if the subjective probability of the upcoming stimulus is above

the threshold and an erroneous trial otherwise. The trigram model has two points (other than the $(0, 0)$ and $(1, 1)$ points). This is because the trigram model predicts 0.25 probability for the all stimuli for the first two trials in each block and 0.625 probability for the more high probability trigram element in all other trials and 0.125 for the other stimuli.

### Kullback-Leibler divergence

We computed KL-divergence between the ground truth probabilities of the task (1.0 for Pattern and 0.25 for Random trials) and that of the inferred internal model's subjective probabilities. For each trial, we computed:

$$\sum_i p_i \cdot (-log(\hat{p}_i) + log(p_i))$$

where $i$ runs over the possible stimuli. Then, we took the mean of all these KL-divergences over the trials in the test sets for Days 2 and 8 for Fig 6G and 6H.

We did the same computation between the inferred Markov models' subjective probabilities and those of the internal models inferred by CT (y-axis on Fig 6G and 6H). For a proof why KL-divergence can be used as a measure of participants' task performance, see S2 Appendix.

## Supporting information

**S1 Appendix. A brief introduction to infinite Hidden Markov Models.**
(PDF)

**S2 Appendix. Optimal prediction and the LATER model.**
(PDF)

**S1 Fig. Experimental design.** *A* Experimental stimuli and abstract representation used in the paper. *B* Design of the experiment. The experiment consisted of ten sessions, separated by a one-week delay. On Days 1–8, participants performed the ASRT task with sequence 1 throughout 25 blocks (5 epochs) each sessions. On Day 9, an interfering sequence (sequence 2) was introduced. Both sequences were tested on Day 10 with blocks of 5 alternating.
(TIF)

**S2 Fig. Graphical representation of internal model and generative model of behaviour.**
Left: Internal model, generative model of the sequence assumed by the participant. Right: generative model of behaviour.
(TIF)

**S3 Fig. Synthetic data experiment.** *A* We first sampled three versions of synthetic internal models using the original iHMM inference method in [33]. The internal models of the synthetic participants differ in their experience (as how many ASRT trials they had seen)—resulting in an "early", "middle" and "late" model. Then, we generated subjective probability values for each model on a new set of ASRT stimuli (holding the pattern sequence intact). *B* Results of our synthetic data experiment. Performance is measured as the amount of variance in response times ($R^2$). We ran our inference method for 81 synthetic datasets with different parameter settings (*symbols with different colors and shapes*). We use the same number of response times as with the human participants to recover the internal models. *Symbol colours* correspond to the response time standard deviation. The result shows that while the response time prediction may be at a lower level, the latent predictive probabilities can still be inferred with relatively high accuracy. This shows the inference method can recover the latent structure from a generated response time sequence. *C* Predictive performance ($R^2$) of the actual internal model of the

synthetic participant vs the predictive performance of the inferred internal model of the same synthetic participant. The inferred model is evaluated on train data sets (same as on panel *B*). *D* Standard deviations of response times of individuals in the first eight experimental sessions.
(TIF)

**S4 Fig. Comparison of quantiles of the (z-scored) *r* variable in the LATER model.** Quantiles are computed from the response times and predicted subjective probabilities with quantiles of the expected normal distribution for the analysed models (*red*, CT; *green*, ideal observer; *blue*, Markov), also known as QQ-plots. Participant-by-participant shows that the empirical distribution of the *r* parameter on a test set is approximately normal with a few exceptions (see participants 124 131 CT and Ideal Observer models), thus validating model assumptions.
(TIF)

**S5 Fig. Response time distributions.** *A* Response time samples generated from different models and the original Data for participant 119. *B* The density plots of the point clouds in *A*. *C* Predicted response times (mean of maximum a posteriori estimates for each model averaged over the model samples) vs actual response times. Response times outside mean ±3 s.d. are omitted for visual clarity. In contrast with panel *A*, the x coordinates are best predictions rather than random samples, hence their spread is much smaller. *D* Histogram of model predictive performances on Day 8. *9* Box plots of model performance distributions, data same as panel *D*.
(TIF)

**S6 Fig. Predicted response time means vs. measured response time means.** Predictions are on the test set on Day 8 of the experiment grouped by three element sequences for each participant separately (each dot corresponds to one possible three-element sequence). Only those sequences were included which had at least 5 measured correct response times in order to limit the standard error over the measured response time mean. Error bars show 2 s.e.m.
(TIF)

**S7 Fig. Predicting when errors will occur for each participant individually.**
(TIF)

**S8 Fig. Model performances of CT models trained on Day 8 and Day 9 for each individual.**
(TIF)

**S9 Fig. Model performances for all models and all participants individually.**
(TIF)

**S10 Fig. Normalized CT performance as a function of the ideal observer model performance on different days of the experiment.** Dots indicate the performance of the models for different individuals.
(TIF)

**S11 Fig. RT distribution examples of synthetic participants.** Each panel shows distributions with RT model parameters sampled from their respective priors. Distributions are shown as violin plots as a function of predictive probabilities.
(TIF)

## Acknowledgments

Resources for the computational analysis were generously provided by the Wigner Data Center. The authors would like to thank to Máté Lengyel, Peter Dayan and Noémi Éltető for comments on an earlier version of the manuscript.

## Author Contributions

**Conceptualization:** Balázs Török, David G. Nagy, Karolina Janacsek, Dezső Németh, Gergő Orbán.

**Data curation:** Balázs Török, Mariann Kiss, Karolina Janacsek, Dezső Németh.

**Formal analysis:** Balázs Török, David G. Nagy, Gergő Orbán.

**Funding acquisition:** Karolina Janacsek, Dezső Németh, Gergő Orbán.

**Investigation:** Balázs Török, Mariann Kiss, Karolina Janacsek.

**Methodology:** Balázs Török, David G. Nagy, Gergő Orbán.

**Project administration:** Karolina Janacsek, Dezső Németh, Gergő Orbán.

**Resources:** Karolina Janacsek, Dezső Németh, Gergő Orbán.

**Software:** Balázs Török.

**Supervision:** Karolina Janacsek, Dezső Németh, Gergő Orbán.

**Validation:** Balázs Török.

**Visualization:** Balázs Török.

**Writing – original draft:** Balázs Török, Gergő Orbán.

**Writing – review & editing:** Balázs Török, David G. Nagy, Karolina Janacsek, Dezső Németh, Gergő Orbán.

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
