## [Decision Letter · Decision Letter 0]

18 Jan 2022

Dear Mr Nagy,

Thank you very much for submitting your manuscript "Tracking the contribution of inductive bias to individualized internal models" for consideration at PLOS Computational Biology.

As with all papers reviewed by the journal, your manuscript was reviewed by members of the editorial board and by several independent reviewers. In light of the reviews (below this email), we would like to invite the resubmission of a significantly-revised version that takes into account the reviewers' comments.

This is an interesting, well posed and carefully conducted study. The reviewers found the proposed method of inferring the internal model from RT using iHMM is novel and relevant. But at the same time, they all asked for a better clarification and more details in methods and results.

We cannot make any decision about publication until we have seen the revised manuscript and your response to the reviewers' comments. Your revised manuscript is also likely to be sent to reviewers for further evaluation.

Sincerely,

Lusha Zhu, Ph.D.

Associate Editor

PLOS Computational Biology

Samuel Gershman

Deputy Editor

PLOS Computational Biology

Reviewer's Responses to Questions

**Comments to the Authors:**

Reviewer #1: Review of "Tracking the contribution of inductive bias to individualized internal models"

Reviewer: Michael Landy

This is quite a fascinating paper, and I was quite impressed with the ability to infer internal models from reaction-time data alone. I think this paper is a substantial contribution to the literature and well worth publishing. I come to this paper with some background and my own work on sequential effects, but I was unfamiliar with the background on which the work was based (beamforming, iHMM's and how to infer them). As such, I think that this paper would benefit from a bit more of a clear tutorial and clarification, and the bulk of my comments below are about making the paper easier to read for the uninitiated.

Specifics (by line number, mostly):

Title: It's interesting that the title stresses inductive bias, whereas the bulk of the paper is about inferring internal models, and the bit about inductive bias, while quite interesting, only comes up at the very tail end of the Results section. I'd think a title that covers the rest would make more sense. The abstract also only talks about inductive bias in the final sentence, which is appropriate.

Figure 2: The legend's description of 2C doesn't match the figure at all ("dots" versus what else? There are no left vs. right panels. What coloured labels are you referring to?).

155-157: This section (and probably some of the earlier text) leads the reader to think that you'll be studying the online learning of a model, and in particular, I thought at this point that you'd be applying the inference to multiple temporal sections of a session to watch that evolution. I only learned much later that your analysis treats the internal model as if it's stable and fixed within a session and, to the extent that you look at the dynamics of learned internal models, you do so at a much slower time scale (across sessions, i.e., across days). Reading the introduction and thinking about the task, I imagined there would be learning within a session and that you'd somehow track that with your inference method. So, I suggest clueing the reader in on what's coming earlier on in the manuscript, as I was disappointed when I learned that the fit of the model using 10 successive blocks was performed only once per session, and was validated by predicting EARLIER blocks. That seemed particularly weird for session 1, when I would have thought the internal model wouldn't be at all stable during the early blocks, when there was little chance that the internal model would be at all stable. I'm surprised there was no discussion at all about within-session, trial-by-trial learning.

176: "Participant-averaged performance of the ideal observer": At this point in the text, unless the reader goes off and reads the Methods carefully, it's not at all clear to the reader what the t-test is comparing and what is meant by performance. The phrase I've quoted would norrmally mean something about how well the ideal observer performs the task. But, by "performance" you mean how well the ideal observer correlates with the human RT data, i.e., the quality of its predictions. This should be rephrased and clarified at this point in the text.

Table 1: I think it would be worthwhile to clarify where the numbers (5/8, 1/8) come from for the trigram model, i.e., half the time the trigram is applied beginning at a pattern trial and its prediction could be perfect, and half the time its aligned with a random trial and it can't predict at all. Not complicated, but worth pointing out explicitly.

Figure 3: Violin plot: Is CT guaranteed to have better performance here (since it's more flexible), or is the plotted performance from the blocks that weren't fit (the earlier blocks in the session)? Also, in the legend, "mode" -> "model".

190: By "better internal model", do you mean closer to the actual generative model or do you mean closer to the flawed model the participant is actually using? The word "better" is best-suited to the former, while I think you mean the latter.

203: I like this analysis of the error trials. But, if you are interested in the predictions made by observers, why not run a version of the experiment in which some trials require prediction (i.e., the task switches from simple RT to prediction, where the task is to say in advance what color is coming up, then get feedback afterward on that trial)? That would be more informative and wouldn't require you to do the analysis only on rare error trials

228-229: It's not clear what these t-tests are testing/comparing (what the CIs are intervals of, especially since you don't say what the means are here). It's also unclear how participants notice an unsignalled change in sequence statistics fast enough for this to work, although of course it DOES work. As I said, the whole manuscript treats internal models as if they are fixed and stable and, here, as if they are instantaneously swapped in and then stable. That's never discussed overtly nor justified, even though it seems, well, counterintuitive or almost certainly false.

Figure 4, and perhaps others: Minor point, but I read and review papers on my iPad (using iAnnotate) and a whole bunch of 4E, including all the data, was not visible. I'm guessing the figure has "layers" and they fooled my iPad's PDF viewer. In generally, you will likely want to flatten your figures (merge the layers) before final submission; that's safer. The legend states that stars mark significant differences, but I don't see any stars in the figure.

237: There is no Suppl. Fig. S5B

267: S6 -> S8

Figure 5A: Shouldn't the ideal "outperform" the trigram model, because the trigram model can't infer the current phase (i.e., whether the current trial is a random or a pattern trial), whereas the trigram model can't make that distinction? That doesn't mean it will correlate better with human observers, but again it might be interesting to separately analyze those two types of trials. Yes, later on in the manuscript you do something about this distinction.

Figure 6C: What are the datapoints in the lower left? Extrema? Never mentioned in the legend.

Legend Figure 6: "accounted for" in part B is about summing R^2 values, which you justify in the running text (as orthogonal elements). But, I thought model "performance" in these graphs was correlation (i.e., R, not R^2), so strictly they shouldn't sum. Please clarify. Also "advantage of normalized CT performance over the ideal observer model". Shouldn't that be over the ideal observer plus the Markov model?

520: Is the starting phase of the sequence fixed or randomized across blocks? Across sessions?

537: The "correct" HMM is a ring of 8 states, half putting out a uniform distribution and half with deterministic output. You never state this explicitly, although it's intrinsic in your discussion of Figure 5F (which is an awesome figure, although I'm not sure how you pull a single inferred model from the inference, when the inference presumably provides a posterior across possible models). I'd think it would be worthwhile to point out that correct model. Even armed with that model, the ideal observer would start out with a distribution across states, and would only lock in after a few trials even if it knew the model with no model uncertainty.

544: \\phi_{s,y} -> \\phi_{S_t,y}

547: problem in 1 -> problem in Eq. 1

548-550: I found these two sentence obscure and even self-contradictory. Please clarify.

558: "uncertainty about the true model AND ACTUAL STATE OF THE STIMULI": I don't understand that latter phrase. In this paper, there is no visual uncertainty.

572 and nearby: I read the Methods early on just after starting to read Results. And, at this point in the Methods I assumed the pseudocode here was applied to groups of 10 blocks, then applied to a slightly later group of 10 blocks, and so on, to understand the dynamics within a session. It was only much later in my reading that I learned that it was only applied to a single, fixed stretch of 10 blocks per session. It would be good to clarify the approacher earlier.

588: Might be worth saying "countably infinite"

590: Fig. ???

608: "(e.g., one session)". First, this is the first time I learned that you assumed no learning was going on within a session (which is a bizarre assumption, although required to make this feasible). Second, the fitting is only for 10 blocks (40% of a session), so you don't really assume it's fixed for the whole session, although since you apply the estimated model to predict a different bit of the session, I guess you are assuming it's fixed for most of the session.

Equation after 610: This bit of notation can be confusing. First, you switch mid-equation from event "S_t = s_t" to shorthand "s_t", even though they mean the same thing (I'd just use the latter). Second, there is a stray proportionalto symbol at the end of the first line. Third, the notation \\hat{s}^t will, for most readers, make them assume this refers to a point estimate of the state at time t. But, you seem to mean it to be an estimate of the probability that "s_t" is the state (given the current and past stimuli). That's a weird form of notation and will confuse people.

611: "For filtering": Here I betray my outsider status: I'm not sure what "filtering" means in this community, so I was confused here. Also, it's worth reminding the reader what is meant by "even if responses times are not considered...". What you mean is that they aren't used in the performance metrics, but are used for model fitting, right?

Table 2: The column headers here are completely messed up, so I couldn't really check/parse this table. Also, what are all they "[1]"s supposed to mean? Looks like some latex formatting got into the table itself.

616-619: Here, at last, is where you finally explicitly state what you are doing about the dynamics, without previewing earlier or justifying that this is a sensible thing to do.

Table 3: This table also has similar messed-up formatting to Table 2. Is \\alpha_0 the same as \\alpha? Where is \\gamma used? What is the "\\cdot \\varepsilon" doing in the definition of \\hat{\\pi}?

631-635: I do STAN model-fitting in my lab, but you are clearly doing something fancier than I've done, so I can't really parse this paragraph (and am unfamiliar with NUTS ;^)

Equation after 642: Why are you using capital letters for \\phi and y all of a sudden?

Eq. after 659: Yes, I think this is basically the LATER formulation (I've used it in one paper, but didn't go back and check). The one weird thing about this formulation is that theoretically an outlier value of r_n could be negative, which would be a problem. For that matter, it could even be zero or very small (i.e., huge tails on the RT distribution).

679: Running it on a simulated observer is obviously a good thing to do. But, did the analysis recover the model that the simulated observer was using? How would you know? How would you score how well it did at recovering the internal model?

686: "below the threshold...": What was that threshold and how was it determined?

733: It's only in this line that you talk about explicitly the issue of models changing under the hood in early trials of a session.

742 et seq.: I was confused by this section and by the corrresponding Results text. First, in this section it wasn't clear what was meant by an error (a wrong button press, of course). It would have been nice to clarify up front that here would would ask whether such finger errors relate to the posterior probabilities of each possible stimulus, both in terms of missing the correct button because it's less likely, and in terms of which button you hit instead, because its probability is high. The main Results text didn't make this clear either. Finally, the ROC idea really is kind of nonsensical, i.e., it doesn't relate clearly as a process model of how finger errors are made.

S2A: The Table in the lower-left of this panel might as well have actual Greek letters as the row headings ;^)

S2B: Again, the labels on the axes should be explained. Is this an R or an R^2? In the legend description of A it says "symbols with different colors and shapes", but this is in panel B, not A.

S6: Are there 64 points in each plot corresponding to the different trigrams?

Reviewer #2: The authors investigated an implicit visuomotor sequence learning task and developed a computational method to reverse-engineer participants’ internal models of the serial dependence through their reaction times (RTs). One major novelty of their method was the use of the infinite hidden Markov model (iHMM) to capture the potentially infinite space of serial patterns that participants might acquire. The authors found that, in explaining participants’ variation in response times, this iHMM model (which was called the CT model in the paper) outperformed both the ideal observer model that follows the ground-truth transition rules and a few Markov models that only tracks the transitions between observable states. The explaining powers of different models changed over the training process, with the earlier stage better approximated by a first-order Markov model and the later stages better by the ideal observer model and a second-order Markov model (i.e., the trigram model). The authors concluded that the failure of achieving an internal model of ground truth as well as its individual differences resulted from specific inductive biases, in particular, a prior belief in first-order Markov transitions.

I think the application of iHMM to modeling human participants’ internal models is novel and insightful. The work is also technically solid. The writing is overall elegant and clear.

But I also have some concerns about the major conclusions of the paper, which I shall specify below.

Major concerns:

1. What parameters of the CT model characterize individual participants’ inductive biases (prior beliefs)? In the CT model, participants were assumed to update their prior beliefs from time to time in a Bayesian way. The deviation of their behaviors from the ideal observer’s depends on their priors. Before reading through the paper, I had thought it would be the hyper-parameters that differed between different individual participants. But when I came to the Methods section, I found the same set of hyper-parameters were used for modeling all participants’ internal models. Then what contributes to the individual differences in the learned internal models (e.g., Fig. 5F)? Did the individual differences just reflect some random variations in participants’ Bayesian inference? Or, did I miss anything?

2. The authors had concluded that the (first-order) Markov model is part of participants’ inductive biases. I was wondering how this conclusion could be compatible with the best-fitting model—the CT model (i.e., iHMM internal model). Links should be made between the internal model predicted by the CT model at early training stages and those of the Markov model. For example, Markov-like internal models might be the emergent properties of iHMM after limited learning experience. Moreover, if so, could it still be claimed that the Markov model constitutes participants’ inductive biases?

3. The authors had shown that the trigram model, a model with the second-order serial dependence, could not explain specific features in participants’ RTs (Fig. 6E). But how about a “quadgram” model with the third-order serial dependence? Considering that it only involves 4*4*4*4 = 256 possibilities, while there were 85 trials/block * 25 blocks = 2125 trials per session and a total of 8 training sessions, it is not a crazy idea that participants might acquire the third-order serial dependence. Besides, such quadgram model seems to be more computationally tractable than iHMM.

4. Could there be any model-free plots and descriptions of the RT results?

5. Working memory capacity had been measured for each participant. Did it correlate with participants’ task performances or their internal models?

Minor issues:

1. I agree that “Cognitive Tomography” (CT) is a cool term. However, I do not think the “CT model” is an appropriate term for the specific CT model based on iHMM, because all the other models in the paper share the same CT framework (internal model + response model). Something like the “iHMM model” might be better.

2. Lines 208–214 and Figure 4A: It seems meaningless to compare the model-predicted proportion of top rank to the chance level. If one model has an overall higher accuracy to predict the incoming stimulus than the other model, its proportion of top rank would be naturally higher than the latter for both correct and incorrect responses. What matters is the discriminability of the proportion of top rank between correct and incorrect responses, such as the ROC reported in Figure 4C.

Line 212: “However, it also assigned the highest probability to the upcoming stimulus in incorrect trials (0.315, n = 2777, p = 1). ” The statistics in the parentheses do not seem to support the statement.

3. I could not quite understand what Figure 4E could tell us. It seems that every model (not necessarily the CT model) could have better predictions for participants’ behaviors when the test statistics were more similar to the training statistics.

I felt even puzzled when I came to Line 616, which reads “throughout the execution of the task, the internal model of the participants is continually updating.” If the updating modeled by the CT model was close to participants’ actual updating, shouldn’t we see similar model performance (R^2) in different test sessions, no matter whether the test statistics were similar to the training statistics or not?

4. Some important details about modeling fitting and comparison methods should be made more explicit in the main text. For example, (1) whether each session of each participant was fitted separately, (2) whether cross-validation was used, and (3) for cross-validation, which parts of data were used as the training set and which as the test set.

Some of these details seem to be described in Table 2, but Table 2 is mis-placed in format and hard to follow.

Line 565: “ten consecutive blocks of trials”. Why ten blocks? Weren’t there 25 blocks in each session?

Line 572: “For each of the 60 posterior model samples”. What does the “60” mean?

5. Line 567: “response times smaller than 180 msecs in each block removed”. What percent of trials were removed?

6. Line 7: “intuitive psychology” seems to be rarely used in the literature. The term “folk psychology” is more common.

7. Finding participants’ internal models to deviate from an ideal observer does not seem to be new or surprising. Why were there so many figures devoted to the comparison between the CT model and the ideal observer model? A comparison between all the models (such as Fig. 5) is more informative and might be better to be described earlier in the paper.

8. Could there be a graphical illustration for the ideal observer model, similar to Fig. 5F? Maybe by enhancing Table 1.

9. The legends of Fig 5D–5F are a little confusing. When I first read “Participant 102 finds a partially accurate model by Day 2 (D) and a model close to the true model by Day 8 (F)”, I had thought (D) and (F) were only about Participant 102.

10. Typos:

Line 221: a space is missing between “model” and “as”.

Line 590: The figure number in the parentheses is missing.

Table 2: The headings of the table seem to be mis-placed.

Reviewer #3: The manuscript presents a new method for inferring subjects' internal models from response times in a sequence learning task. Because the task's statistical structure is unknown to the subjects, the assumption that the subject's internal model matches the true generative model of the task (the ideal observer assumption) does not hold. Thus, the authors use a flexible class of dynamical models (iHMM) to represent subjects' internal models and combine it with a behavioral model linking subjective probabilities to response times. The iHMM-based internal models estimated from response time data are shown to predict response times better compared to the ideal observer. By considering an alternative model, which assumes no hidden structure but only Markovian dependencies, the authors show that all subjects start with a bias towards simple temporal dependencies, but some subjects learn a model closer to the ideal observer.

The conceptual introduction to the problem is very clear. Particularly, the explicit distinction between the subject's internal model and the behavioral model accompanied by the graphical model notation (Fig. S2) is quite helpful. The results are interesting and the "cognitive tomography" method constitutes a relevant contribution to the recent literature on inferring internal models from behavioral data. However, the description of the methods could be improved in terms of clarity and level of detail and some aspects of the results need clarifying statements or additional analyses:

- Clarifying what exactly constitutes a state of the dynamical system in the article's main text would help readers not well-versed in HMMs and similar models. Relatedly, while the graphical notation for the model (Fig. 2A) is very informative once understood, it is worth a little more explanation: e.g., the authors could show how the true dynamics of the task (i.e. the ideal observer's model) or an instance of the Markov model look in the graphical notation, which would make it easier for the reader to appreciate the inferred models (Fig. 5D,F).

- The validation of the inference method on synthetic datasets is a bit scarce. For a model of this complexity and a highly customized inference procedure, I would expect to see some prior predictive checks, inference diagnostics (e.g. r_hat, effective sample size), and posterior predictive checks. As a guideline for reporting Bayesian analyses, I suggest Kruschke (2021).

- The evaluation presented in Fig. S3B suggests that the response time prediction performance is not particularly good. Even for synthetic datasets with response time standard deviations comparable to real data (the darker dots), the R^2 values are mostly between 0.25 and 0.75. Is this just the result of the inherent variability in the response times due to the LATER model (which the better performance in predicting subjective probabilities might suggest) or is this due to a failure of the inference method? Could one compare the response time prediction performance against an upper bound on the response time prediction performance computed from the ground truth synthetic internal model? How well are the parameters of the response time model (tau, mu, sigma) recovered by the inference method?

- The model comparison based on R^2 is not really convincing, because it does not take into account the significantly higher model complexity of the iHMM-based model. The authors chose R^2 because not all models are Bayesian. To my understanding, the only non-Bayesian model is the trigram model, which is not central to the argument in the paper, while the ideal observer, the iHMM-based model, and the Markov model are Bayesian. If this is correct, the authors should perform a Bayesian model comparison for these three models. If this is incorrect, please expand the description of the models in the paper to make clearer how they are fit to the data.

- In the Section "Trade-off between ideal observer and Markov model contributions", the inferred internal model is only compared quite indirectly to the ideal observer model, via their predictive accuracy for response times. Is there a more direct way to assess the distance between an iHMM and the true HMM employed in the experiment? The evaluations presented in the original iHMM paper by Gael et al. (2008) suggest that there is.

- I agree with the point made in the discussion, that POMDPs are a possible alternative formalization for the problem at hand. While the authors acknowledge in the introduction that the internal model of the agent need not be identical to the true generative model of the task, this point also applies here: An agent might assume that their actions influence the state, while it is actually not the case in the true generative model. Furthermore, employing a POMDP formulation might also shed light on internal costs relevant to the task (e.g. computational costs), which are absent from HMMs without explicit modeling of actions. I think these points are worth further discussion.

Minor points:

- The argument in the introduction for moving beyond ideal observers could be strengthened further by including relevant literature making similar arguments (e.g. Feldman, 2013, Beck et al., 2012).

- "learning a novel statistics" should be "learning novel statistics" (p. 3, l. 22), same on p. 4, l. 58, p. 10 l. 200, 201, 222

- Fig. S3A refers to Gael (2011). Should this be Gael et al (2008) as in the caption and in the bibliography or is it referring to a different paper?

- "and we formulated as a trigram model" (grammar and meaning??)

- Fig. 5B is not referenced

- Latex: use $M_\\text{education}$ (\\text environment for whole words) instead of $M_{education}$

- p. 22 l. 590: missing figure reference

- Table 2: headings seem to be broken

- p. 14: spelling of normalized / normalised is inconsistent

**Have the authors made all data and (if applicable) computational code underlying the findings in their manuscript fully available?**

Reviewer #1: Yes

Reviewer #2: None

Reviewer #3: **No: **While zip files containing data and code were available, the passwords for these files were only available upon request from the authors.

PLOS authors have the option to publish the peer review history of their article (what does this mean?). If published, this will include your full peer review and any attached files.

Reviewer #1: **Yes: **Michael S Landy

Reviewer #2: No

Reviewer #3: No
---

## [Decision Letter · Decision Letter 1]

4 Apr 2022

Dear Mr Nagy,

Thank you very much for submitting your manuscript "Tracking the contribution of inductive bias to individualized internal models" for consideration at PLOS Computational Biology. As with all papers reviewed by the journal, your manuscript was reviewed by members of the editorial board and by several independent reviewers. The reviewers appreciated the attention to an important topic. Based on the reviews, we are likely to accept this manuscript for publication, providing that you modify the manuscript according to the review recommendations.

The revised manuscript has addressed most issues raised by referees. I'm returning the manuscript to you to address a few minor comments from two of the reviewers.

Sincerely,

Lusha Zhu, Ph.D.

Associate Editor

PLOS Computational Biology

Samuel Gershman

Deputy Editor

PLOS Computational Biology

[LINK]

The revised manuscript has addressed most issues raised by referees. I'm returning the manuscript to you to address a few minor comments from two of the reviewers.

Reviewer's Responses to Questions

**Comments to the Authors:**

Reviewer #1: Re-review of "Tracking the contribution of inductive bias to individualized internal models"

Reviewer: Michael Landy

I was impressed with this paper the first time, but needed a bit more clarity in the presentation. This version improves that quite a bit and adequately responds to my and the other reviewers' requests (IMHO). My comments are pretty minor.

Specifics:

lines 99-100: This says that sessions were on consecutive days, but the Procedure says they were spaced a week apart.

Figure 3: The diagram on the lower right is probably supposed to be the Markove model, but it is neither mentioned nor described at all in the legend.

276-277: "successfully recruit previously learned models": I thought this referred to the fact that on Day 10 the better-performing model tracks the switching. That's clear in Fig. 5E, but is not exactly convincing across subjects glancing as Fig. S8 and no summary across subjects is provided.

294: vanes -> wanes

Fig. 6D,F: The correspondence between the model samples and participants/days is not clarified in the legend and is indicated by red circles on day numbers and a skinny red line connecting the model to that day that I missed completely until I stared at the figure for quite some time. You should make the connections more obvious AND mention it in the legend. Another option (not mutually exclusive) is to put a title in the corner of each model sample that says something like "Participant 102, Day 2".

327: The main text here only suggests that KL is used to measure the match between predictions, whereas the Appendix gives another justification for why KL is the right thing to do. You should allude to that other justification here as well.

338: identified FOR all?

340: FOR several participants (or maybe IN, but not AT)

436: characteristic OF all participants

3 after 603: A glitch here. S_t,Y_t \\memberof \\mathcal{N}. The sequence... [Yeah, I know my latex is wrong ;^)]

695: inbetween -> in between

697: last 60 -> the last 60

Table 1, Hierarchical prior over state transitions, Notation: Shouldn't that be \\alpha,\\gamma ???

711: predictions many -> prediction of many

759: seen on -> seen in

775: are subset -> are a subset

822: take THE mean ... over THE last 60

824: IN Fig. 5

827: predicting A correct trial

Supplement, p. 2, para. 2: The reference for Hierarchical Dirichlet didn't get filled in.

Supplement, Fig. S3: I have a notation here "Compare the resulting HMMs". I can't remember what sort of comparison I was thinking of... ;^(

Appendix B, para. 2: Note that THE quantity we obtained...

Reviewer #2: I have no further questions.

Reviewer #3: I appreciate the effort the authors have made to address my previous comments and I think the clarity of the manuscript is improved. Specifically, the description of the iHMM is now more detailed and accessible for uninitiated audiences. My concerns about the model validation and comparison have been resolved by the addition of the new version of Fig. S3B and the clarified description of how the models were fitted and evaluated. With most methodological concerns out of the way, I only have one remaining minor issue: In my previous review, I asked for MCMC diagnostics because of the custom inference method and high model complexity. While the authors have pointed to a plot validating an assumption of the model and clarified how much data were used to fit the model, they have not shown evidence for the convergence of the MCMC method.

**Have the authors made all data and (if applicable) computational code underlying the findings in their manuscript fully available?**

Reviewer #1: **No: **I can't find any mention of it in the main text nor the supplement

Reviewer #2: None

Reviewer #3: Yes

PLOS authors have the option to publish the peer review history of their article (what does this mean?). If published, this will include your full peer review and any attached files.

Reviewer #1: **Yes: **Michael S Landy

Reviewer #2: No

Reviewer #3: No

Figure Files:

Data Requirements:

Reproducibility:

References:

---

## [Decision Letter · Decision Letter 2]

8 May 2022

Dear Mr Nagy,

We are pleased to inform you that your manuscript 'Tracking the contribution of inductive bias to individualized internal models' has been provisionally accepted for publication in PLOS Computational Biology.

Best regards,

Lusha Zhu, Ph.D.

Associate Editor

PLOS Computational Biology

Samuel Gershman

Deputy Editor

PLOS Computational Biology

Reviewer's Responses to Questions

**Comments to the Authors:**

Reviewer #1: All my comments have been adequately addressed, since they were almost all trivial! I didn't check those about the supplement...

Reviewer #3: Thanks for providing evidence for the convergence of their MCMC chains by showing the posterior CIs across multiple chains. I am still not quite sure why standard diagnostic method for checking MCMC like R-hat or effective sample size were not provided.

Irrespective of this minor issue, I think this is a very interesting paper and well worth publishing.

**Have the authors made all data and (if applicable) computational code underlying the findings in their manuscript fully available?**

Reviewer #1: Yes

Reviewer #3: Yes

PLOS authors have the option to publish the peer review history of their article (what does this mean?). If published, this will include your full peer review and any attached files.

Reviewer #1: **Yes: **Michael S Landy

Reviewer #3: No

---

## [Editor Report · Acceptance letter]

10 Jun 2022

PCOMPBIOL-D-21-02227R2 

Tracking the contribution of inductive bias to individualized internal models

Dear Dr Nagy,

I am pleased to inform you that your manuscript has been formally accepted for publication in PLOS Computational Biology. Your manuscript is now with our production department and you will be notified of the publication date in due course.

With kind regards,

Anita Estes
